# A LoD of Gaussians: Unified Training and Rendering for Ultra-Large Scale Reconstruction with External Memory

## Abstract

Gaussian Splatting has emerged as a high-performance technique for novel view synthesis, enabling real-time rendering and high-quality reconstruction of small scenes. However, scaling to larger environments has so far relied on partitioning the scene into chunks—a strategy that introduces artifacts at chunk boundaries, complicates training across varying scales, and is poorly suited to unstructured scenarios such as city-scale flyovers combined with street-level views. Moreover, rendering remains fundamentally limited by GPU memory, as all visible chunks must reside in VRAM simultaneously. We introduce *A LoD of Gaussians*, a framework for training and rendering ultra-large-scale Gaussian scenes on a single consumer-grade GPU without partitioning. Our method stores the full scene out-of-core (e.g., in CPU memory) and trains a Level-of-Detail (LoD) representation directly, dynamically streaming only the relevant Gaussians. A hybrid data structure combining Gaussian hierarchies with Sequential Point Trees enables efficient, view-dependent LoD selection, while a lightweight caching and view scheduling system exploits temporal coherence to minimize the loading overhead. Together, these innovations enable seamless multi-scale reconstruction and interactive visualization of complex scenes—from broad aerial views to fine-grained ground-level details.

## 1 Introduction

Given a set of posed images of a 3D scene, the task of novel view synthesis (NVS) is to generate plausible images of the scene from unseen viewpoints. Early approaches achieved this via image-based blending (Zhang et al., 2019), but the introduction of Neural Radiance Fields (NeRF) (Mildenhall et al., 2020) marked a breakthrough, enabling high-quality results by optimizing an implicit volumetric scene representation through a multi-layer perceptron. More recently, 3D Gaussian Splatting (3DGS) (Kerbl et al., 2023) extended this paradigm to an explicit representation: a set of Gaussian primitives that are efficiently rasterized using splatting techniques (Zwicker et al., 2001), replacing costly ray marching and allowing real-time rendering with fast convergence.

Despite these advances, both NeRF and 3DGS remain constrained by memory bottlenecks when applied to large-scale environments. Prior methods address this by dividing scenes into smaller chunks (Xu et al., 2023; Tancik et al., 2022; Kerbl et al., 2024; Liu et al., 2024; Chen et al., 2024; Lin et al., 2024), training each independently before merging results. While chunking strategies mitigate memory usage during training, they introduce several key limitations:

1. **View-chunk misalignment:** Camera views often span multiple chunks, especially in open environments or multi-scale datasets (e.g., combining aerial and street-level images). This makes chunk boundaries arbitrary and hard to define, complicating scene partitioning and training. We identify two artifacts caused by view-chunk misalignment: *Chunk bleeding* occurs when Gaussians extend out of their current chunk and obscure neighbouring chunks after merging. When an occluder on a training image is not part of the current chunk, its "ghostly" outline will be trained into the wrong chunk, leading to *Chunk ghosting*. Examples of these artifacts can be seen in Figures 7 and 11.

2. **Redundant overlap:** To avoid artifacts at chunk boundaries, regions are typically trained with significant overlap, which increases memory usage and prolongs training time.

3. **Asymmetric hardware demands:** Although chunking reduces memory requirements during training, rendering may require all visible chunks in memory simultaneously—often exceeding the capacity of the original training setup.

The simplest and most robust alternative to chunking is to avoid splitting altogether. With *A LoD of Gaussians*, we introduce a seamless pipeline that enables training and rendering of ultra-large-scale scenes directly on a single consumer-grade GPU, without any form of scene partitioning (see Figure 8). To handle scenes that exceed available VRAM, we store all Gaussian data in CPU RAM and dynamically stream only those visible from the current training view into GPU memory. Still, a single distant view could require access to the full scene. To address this, we construct a hierarchical Level-of-Detail (LoD) model inspired by Kerbl et al. (2024), loading detail proportional to view distance. Maintaining this hierarchy during training is non-trivial, as Gaussian properties evolve dynamically. We propose a novel hierarchy densification strategy, adapted from the MCMC-style spawning (Kheradmand et al., 2024), to support stable, progressive refinement. Efficient view-dependent selection from the hierarchy is challenging for large models. Instead of full tree traversal, we adopt Sequential Point Trees (SPTs) (Dachsbacher et al., 2003), originally developed for point cloud LoD rendering. Our Hierarchical SPT version allows us to compute the correct LoD cut efficiently for rendering individual views and camera paths. Finally, to reduce CPU-GPU data transfer overhead, we introduce a lightweight caching system that tracks recently used Gaussians and reuses them across training iterations. In summary, we make the following contributions:

1. We propose a novel hierarchy densification strategy that enables dynamic expansion and restructuring of Gaussian LoD representations during training.

2. We leverage the Sequential Point Tree (SPT) data structure and adapt it for large-scale Gaussian Splatting to perform fast, parallelizable LoD cuts for efficient training and rendering.

3. We demonstrate, for the first time, how external memory can be used to train and render ultra-large Gaussian scenes seamlessly on consumer-grade GPUs.

4. We design a caching and view scheduling system that exploits temporal coherence to minimize data transfer overhead and improve training throughput.

5. We verify trough extensive evaluations that the pipeline succeeds in reconstructing huge datasets with complex camera distribution, where SOTA divide-and-conquer methods fail even with larger VRAM budgets.

## 2 RELATED WORK

**Large Scale Reconstruction**    Reconstructing large-scale scenes from images has long been a central challenge in visual computing. Traditional approaches relied on Structure-from-Motion (SfM) pipelines to recover geometry from unordered photo collections (Agarwal et al., 2009; Schönberger & Frahm, 2016). Differentiable rendering techniques, notably NeRF (Mildenhall et al., 2020) and 3DGS (Kerbl et al., 2023), marked a paradigm shift by optimizing volumetric scene representations. Extensions of NeRF to large scenes typically employ scene partitioning (Tancik et al., 2022; Xu et al., 2023) or multi-GPU training strategies (Li et al., 2024b). Similarly, most large-scale 3DGS pipelines adopt chunk-based training: *Hierarchical-3DGS* (H-3DGS) (Kerbl et al., 2024) trains chunks independently and then merges them into a global LoD hierarchy; *CityGaussian* (Liu et al., 2024) combines chunked training with per-chunk LoD selection using *LightGaussian* (Fan et al., 2024); *OccluGaussian* (Fan et al., 2024) partitions the scene to maximize camera correlation in each chunk; and *VastGaussian* (Lin et al., 2024) introduces decoupled appearance modeling and progressive partitioning. *Horizon-GS* (Jiang et al., 2024) integrates divide-and-conquer strategies with LoD mechanisms from Ren et al. (2024), specifically targeting hybrid aerial/street-view datasets. *GrendelGS* (Zhao et al., 2024) avoids spatial chunking by distributing training images across GPUs, such that each device renders a disjoint screen region. Another research direction focuses on extracting geometric proxies from large-scale 3DGS scenes (Li et al., 2025; Liu et al., 2025; Chen et al., 2024). These methods leverage TSDF fusion and geometric losses to generate multiple meshes, which are fused and rendered efficiently using traditional rasterization.

**Level-of-Detail Rendering**    Level-of-detail techniques reduce the geometric complexity of distant scene content to accelerate rendering. In the context of 3DGS, LoD approaches have been explored for

enabling efficient rendering on memory-constrained or mobile devices. Compression-based strategies include attribute quantization via codebooks, pruning of low-impact Gaussians, and adapting the degree of spherical harmonics per primitive (Papantonakis et al., 2024; Niedermayr et al., 2024; Fan et al., 2024; Fang & Wang, 2024; Huang et al., 2025; Niemeyer et al., 2025; Seo et al., 2024). *Scaffold-GS* (Lu et al., 2023) uses latent vectors anchored to reference Gaussians, with an MLP generating associated Gaussians at render time. *Octree-GS* (Ren et al., 2024) extends this concept to hierarchical LoD rendering, enabling real-time control over detail levels through spatial subdivision. *Virtualized 3D Gaussians* (Yang et al., 2025) focuses on rendering composed scenes from individual reconstructed objects using an LoD approach inspired by Unreal Engine 5's Nanite (Karis et al., 2021).

## 3 PRELIMINARIES

Here, we briefly review 3DGS and how Gaussian hierarchies enable adaptive level-of-detail rendering.

### 3.1 HIERARCHICAL 3D GAUSSIAN SPLATTING

3DGS (Kerbl et al., 2023) models a radiance field using a set of spatially distributed Gaussians, each with mean $\boldsymbol{\mu}_i \in \mathbb{R}^3$, RGB base colors $\mathbf{b}_i \in \mathbb{R}^3$ and covariance matrices $\boldsymbol{\Sigma}_i = \mathbf{R}_i \mathbf{S}_i \mathbf{S}_i^\top \mathbf{R}_i^\top$, which are parameterized via a diagonal scaling matrix $\mathbf{S}_i = \operatorname{diag}(s_i^1, s_i^2, s_i^3)$ and an orthonormal rotation matrix $\mathbf{R}_i$. Each Gaussian also stores an opacity $\sigma_i$ and view-dependent color, modeled using spherical harmonics (SH) coefficients $\mathbf{f}_i^d$. The SH degree $d$ controls expressiveness, with each Gaussian requiring $\sum_{j=1}^d 3 \cdot (2j+1)$ parameters. For rendering, all $N$ Gaussians are sorted by distance to the camera and a discrete approximation of the volume rendering equation is evaluated for every pixel $\mathbf{x}$ with corresponding view direction $\mathbf{v}$:

$$\mathbf{C}(\mathbf{x}) = \sum_{i=1}^N \mathbf{c}_i(\mathbf{v}) \alpha_i(\mathbf{x}) \prod_{j=1}^{i-1} (1 - \alpha_j(\mathbf{x})), \tag{1}$$

where $\alpha_j$ is the opacity of the $j$-th Gaussian along the view ray:

$$\alpha_j(\mathbf{x}) = \sigma_j e^{-\frac{1}{2}(\mathbf{x} - \boldsymbol{\mu}_j')\boldsymbol{\Sigma}'(\mathbf{x} - \boldsymbol{\mu}_j')^T}. \tag{2}$$

Here $\boldsymbol{\mu}'$ and $\boldsymbol{\Sigma}'$ denote the projected 2D mean and covariance on the image plane, obtained by applying an affine approximation of the projective transform (Zwicker et al., 2001).

**3DGS Memory** Standard 3DGS pipelines store the full set of per-Gaussian attributes, training images, and optimizer state in GPU memory (VRAM); see Figure 14 for a detailed per-Gaussian breakdown. Additional temporary allocations occur during forward and backward passes (e.g., for sorting and gradient accumulation). This overhead varies with the scale of the Gaussians and effectiveness of culling strategies, but can be roughly upper-bounded by around 800 bytes per Gaussian in practice. This limits typical training to roughly $500\,000$ Gaussians per GB of GPU memory, imposing strict constraints on the detail and extent of reconstructions.

**Gaussian hierarchies** as introduced in H-3DGS (Kerbl et al., 2024), recursively merge nearby Gaussians into a tree, where each non-leaf node approximates its children, and leaves correspond to the original Gaussians. A cut is defined by a condition $c_{\text{hier}}(i, \text{cam})$ evaluated in a breadth-first search (BFS). If a node satisfies the cut condition, it is added to the cut set and its children are skipped; otherwise, the BFS continues. A *proper cut set* includes no parents or children of any included node and thus provides a view-adaptive LoD representation.

The cut condition used in Kerbl et al. (2024) is a simple cutoff to the camera distance:

$$c_{\text{hier}}(i, \text{cam}) = \|\boldsymbol{\mu}_i - \mathbf{p}_{\text{cam}}\|_2 \geq m_d(i), \quad m_d(i) = \frac{T}{\max_j s_i^j}, \tag{3}$$

where $\mathbf{p}_{cam}$ is the camera position, $T$ is a global LoD threshold, and $m_d(i)$ is the minimum acceptable distance for viewing Gaussian $i$. The BFS ensures that if $i$ is in the cut set, parent($i$) failed the condition: $m_d(\text{parent}(i)) > \|\boldsymbol{\mu}_i - \mathbf{p}_{\text{cam}}\|_2 \geq m_d(i)$.

# 4 METHOD

To train models that exceed GPU memory limits, all Gaussian attributes are stored in CPU RAM and streamed to the GPU on demand for each training view according to the LoD hierarchy. To accelerate the hierarchy cut, we store a copy of only the tree structure in VRAM, where larger subtrees are replaced by *Sequential Point Trees* (SPTs), forming a *hierarchical SPT* (*HSPT*). To minimize costly transfers between RAM and VRAM, we track which SPTs currently reside in GPU memory and at which detail. Only if an SPT is not present in this GPU cache at a similar level of detail, will it be loaded from RAM. Densification is performed on the CPU by adding new leaf nodes to the hierarchy and respawning low-opacity leaf nodes. The densified hierarchy is then converted back to an HSPT and transferred to the GPU for a new round of training iterations. Appendix A.2 includes more details on initialization and training. An overview of our training and densification process can be found in Figure 1 and pseudocode in Appendix A.8.

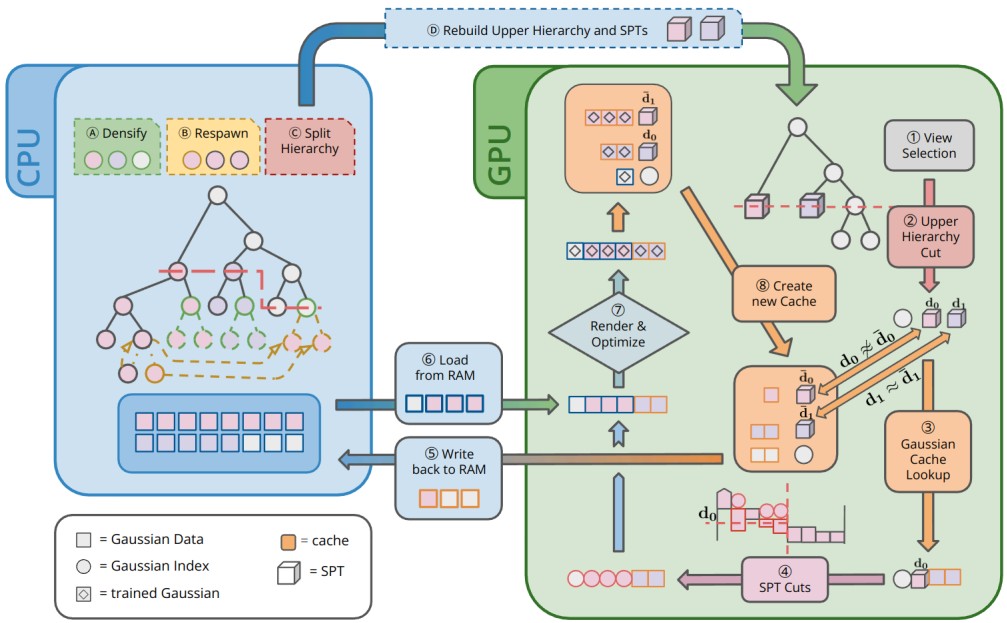

Figure 1: Method Overview: Steps ① to ⑧ show the process of a single training iteration, while Ⓐ through Ⓓ show a densification step.

## 4.1 SEQUENTIAL POINT TREES FOR GAUSSIAN SPLATTING

Sequential Point Trees (Dachsbacher et al., 2003) were originally developed for point cloud LoD rendering, but can be trivially extended to ellipsoids and Gaussians. They enforce a more constrained but more efficient cut condition than Kerbl et al. (2024):

$$c_{\text{SPT}}(i, \text{cam}) = m_d(\text{parent}(i)) > \|\boldsymbol{\mu}_{\text{root}} - \mathbf{p}_{\text{cam}}\|_2 \geq m_d(i). \quad (4)$$

This condition is evaluated for all Gaussians in parallel, using the shared root-camera distance $\|\boldsymbol{\mu}_{\text{root}} - \mathbf{p}_{\text{cam}}\|_2$. It requires storing only sorted pairs $(m_d(i), m_d(\text{parent}(i)))$, significantly reducing memory compared to full Gaussian hierarchies. To optimize cuts, Gaussians are sorted by $m_d(\text{parent}(i))$ in descending order. A binary search determines the cutoff index $N$, above which Gaussians are too fine to be rendered. Note that cuts are guaranteed to be proper, with nodes where $m_d(\text{parent}(i)) > m_d(i)$ never being selected for the cut.

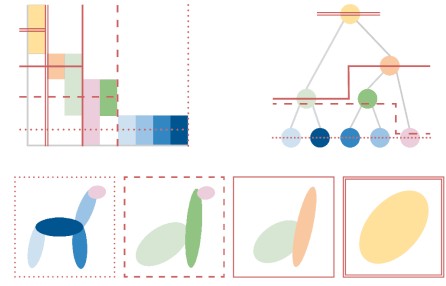

Figure 2: An SPT and Gaussian hierarchy represent the same 5 Gaussians at varying levels of detail shown with four possible hierarchy cuts in red. Vertical lines in the SPT show the binary search result and horizontal lines the distance cut.

Evidently, the level of detail of all Gaussians in the SPT is dictated by the camera's distance to its root node. This can lead to Gaussians with $m_d(i) > \|\boldsymbol{\mu}_i - \mathbf{p}_{cam}\|_2$ being rendered, even though they would be too coarse for the current view. To counteract this issue, we define: $M_d(i) = m_d(i) + \|\boldsymbol{\mu}_i - \mathbf{p}_{cam}\|_2$, as a conservative minimum distance function. By the triangle inequality, selecting Gaussians satisfying $M_d(i) \leq \|\boldsymbol{\mu}_{\text{root}} - \mathbf{p}_{\text{cam}}\|_2$ guarantees $m_d(i) \leq \|\boldsymbol{\mu}_i - \mathbf{p}_{\text{cam}}\|_2$. In turn, this means that Gaussians that are further away from the camera than the root node will be selected at a higher level of detail. SPTs are best suited for tightly grouped Gaussians observed from distances greater than their mutual spacing. Their compact memory footprint and parallel evaluation make them well-suited for large-scale scenes. Figure 2 visualizes both hierarchy types and their LoD cuts for a toy example.

## 4.2 DENSIFICATION

Densifying an LoD representation presents a unique challenge, as the hierarchical structure must evolve continuously during training. Prior works circumvent this issue by constructing LoD hierarchies only after chunk-level training and densification are complete.

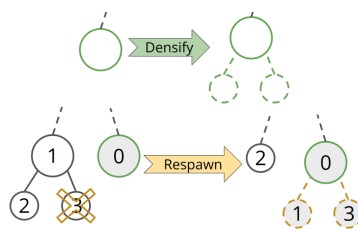

We take inspiration from *3DGS-MCMC* (Kheradmand et al., 2024), which 'splits' Gaussians, replacing them with two new Gaussians which together should appear similarly to the original. Notably, this mirrors how a parent node in a Gaussian hierarchy approximates its children. Therefore, we adopt this approach and instead 'spawn' two new child nodes for a Gaussian, increasing the size of the hierarchy with minimal artifacts.

Figure 3: Example of densifying and respawning leaf nodes.

Instead of pruning, Kheradmand et al. (2024) declares Gaussians below a certain opacity threshold as 'dead', and respawns them at the position of a high-opacity Gaussian. We propose a similar strategy: when a leaf node dies, its parent is replaced by its sibling node; the dead leaf node and its parent are then respawned as children to another node, which is selected to be densified. See Figure 3 for an overview of the two hierarchy densification operations. Together, they ensure that the hierarchy can be expanded during training in a stable manner and rebalanced as required.

While Kheradmand et al. (2024) choose Gaussians to densify using a random selection weighted by opacity, we employ the strategy from Kerbl et al. (2024), which selects Gaussians for densification based on their maximal screen-space gradient. Avoiding destructive opacity resets, as advocated by Kerbl et al. (2023), is particularly important in our setting, as many Gaussians will no longer conform to the hierarchical structure after opacity is restored to normal. Further details on our densification method can be found in Appendix A.7.

## 4.3 THE HIERARCHICAL SPT DATASTRUCTURE

This section covers previous approaches to LoD selection and why a new datastructure—the hierarchical SPT—is necessary for robust and efficient training.

**BFS**  Computing the cut set of a large Gaussian hierarchy is a costly operation that must be performed for every frame. A straightforward solution is to run a breadth-first search (BFS) from the root. This guarantees a proper cut and enables early pruning of large subtrees, e.g., via frustum culling. However, graph traversal is not well suited to parallel execution on the GPU.

**Parallel Cut**  To enable GPU-accelerated cuts, Kerbl et al. (2024) evaluate the cut condition in parallel for each Gaussian:

$$\Big(m_d(i) < \|\boldsymbol{\mu}_i - \mathbf{p}_{cam}\|_2\Big) \wedge \Big(m_d(parent(i)) \geq \|\boldsymbol{\mu}_{\text{parent}(i)} - \mathbf{p}_{cam}\|_2\Big), \qquad (5)$$

where any Gaussian that is sufficiently small at its current camera distance and whose parent is too large to be rendered, should be part of the cut set. This produces a proper cut under the assumption that child Gaussians always have a smaller minimal distance than their parents (i.e. the heap condition

is fulfilled): $\forall i : m_d(i) < m_d(parent(i))$. This is generally valid when hierarchies are constructed after training, since parent Gaussians represent coarser approximations of their children. However, when the hierarchy is modified during training and densification, optimization can break the heap condition—leading to invalid cut sets and degenerate hierarchies that worsen over time.

**Hierarchical SPT (HSPT)**  We present the hierarchical SPT data structure, which combines the benefits of both approaches. To construct it from a Gaussian hierarchy, we cut it using a BFS on the condition $c_{\text{HSPT}}(i) = s_i^1 \cdot s_i^2 \cdot s_i^3 < \texttt{size}$ with volume threshold $\texttt{size}$. The resulting cut set $\mathbb{C}_{\text{HSPT}}$ partitions the hierarchy into the *upper hierarchy*, which includes all Gaussians with volume greater than $\texttt{size}$, and the *lower hierarchy*, consisting of the subtrees rooted at the nodes in the cut set.

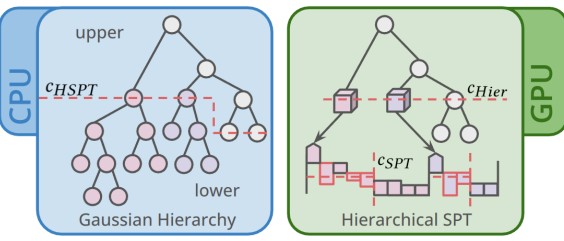

Figure 4: A Gaussian hierarchy is converted to an HSPT by cutting according to Gaussian volume and converting sufficiently large subtrees to SPTs. The HSPT can then be cut in a 2-step process.

The volume of each root node in the lower hierarchy is now bounded by $\texttt{size}$, which also roughly bounds the extent of all Gaussians in the subtree. This provides an upper bound on the error introduced if the subtree is converted into an SPT. Consequently, each subtree of sufficient size in $\mathbb{C}_{\text{HSPT}}$ can be transformed into a Sequential Point Tree to accelerate cut computation. The HSPT-based cutting process then proceeds in two steps: first, a BFS on the upper hierarchy selects the required nodes and leaf/SPT subtrees for the current view. Second, each selected SPT is cut according to the camera's distance to its root node. Together, these yield the full set of Gaussians needed for rendering the current frame. The construction and cutting process of an HSPT is illustrated in Figure 4. Figure 10 and Figure 15 show example frames with respectively SPTs highlighted and different levels of detail.

Rebuilding the HSPT for every training iteration would eliminate any performance benefit. Instead, we exploit the fact that the minimum distance $m_d$ evolves slowly during optimization and thus only needs to be updated infrequently. In practice, we rebuild the HSPT only after each densification step.

This infrequent recomputation allows us to use a more accurate—albeit more expensive—minimum distance metric than the inverse of maximal scale. Specifically, we define:

$$m_d'(i) = \frac{T}{\sqrt{s_i^1 \cdot s_i^2 + s_i^1 \cdot s_i^3 + s_i^2 \cdot s_i^3}}, \tag{6}$$

which corresponds to the inverse square root of the surface area of the Gaussian ellipsoid (up to a constant factor). This better captures the perceived size of anisotropic Gaussians, especially those that are significantly elongated in one or more directions.

**Frustum Culling**  The major benefits of using BFS to cut the upper hierarchy are the guarantee of a proper cut and early culling of subtrees. To this end, we frustum cull every node considered in the BFS by checking if a sphere around the Gaussians with radius equal to $(3 \cdot \max_j s_i^j)$ intersects with the view frustum. Using the Gaussian scale as a proxy for the extent of its entire subtree is not perfectly accurate, but comparisons to using a full bounding sphere hierarchy showed no discernable difference in our experiments. It should be noted that Gaussians are implicitly frustum culled during rasterization, but this early culling accelerates the cutting procedure and significantly decreases the number of Gaussians that need to be loaded from RAM, as can be seen in Figure 9.

## 4.4 CACHING ON THE GPU

Loading Gaussian data from RAM is a costly operation that can become a significant bottleneck during large-scale training. To mitigate this, we maintain a GPU-resident cache of Gaussians that are likely to be reused across consecutive training views. However, checking the cache for every individual Gaussian would introduce non-trivial overhead. Once again, SPTs offer an efficient alternative.

Rather than caching individual Gaussians, we store the Gaussians from SPT cuts along with the cached distance from the camera to the root of each SPT, denoted $\bar{d}^j$ for the $j$-th SPT. During rendering, when the upper hierarchy is cut and the required SPTs identified, we compute $d^j = \|\boldsymbol{\mu}_{\text{root}(j)} - \mathbf{p}_{\text{cam}}\|_2$ and check whether a matching cut is cached, using a simple distance ratio tolerance:

$$D_{\min} \leq \frac{d^j}{\bar{d}^j} \leq D_{\max}. \qquad (7)$$

Here, $D_{\max}$ defines the allowable range for using coarser-than-optimal detail, while $D_{\min}$ limits how much finer detail can be tolerated. If this condition is met, the cached SPT cut is reused, avoiding a costly RAM-to-GPU transfer.

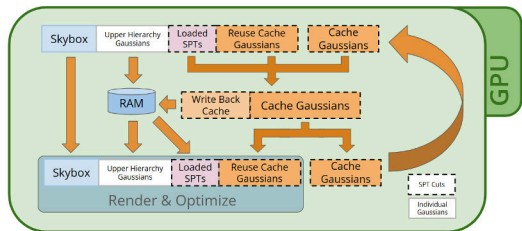

Figure 5: Gaussians required for the current training view are assembled from three sources: the upper tree, newly loaded SPT cuts from RAM, and cache hits. After optimization, newly accessed SPTs are added to the GPU cache.

While this heuristic introduces slight variability in rendered detail—since the LoD may depend on the cache state—we find that this stochasticity actually improves training robustness. In particular, the subtle variation in detail across views discourages overfitting to fixed camera distances and promotes better generalization across scales.

For each training view, visible Gaussians are assembled from the upper hierarchy, the cached SPTs, and the skybox Gaussians (which remains in VRAM). Uncached SPTs are streamed from RAM. After each training iteration, newly loaded SPTs are added to the cache.

To bound VRAM usage, we use a least-recently-used (LRU) write-back policy. When a memory threshold is exceeded, entries are written back to RAM. Additionally, to prevent overfitting to persistent cache entries, the entire cache is flushed every $1\,000$ iterations. Figure 5 illustrates the caching process across two frames.

**View Selection** In large-scale scenes, the GPU cache typically covers only a small fraction of the overall geometry, leading to sparse cache hits. To improve cache utilization, we prioritize spatial locality by selecting successive training views close to the current one, maximizing Gaussian reuse.

To this end, we precompute a $k$-nearest-neighbour graph over all training view positions, where edge weights $w_{ij}$ correspond to the Euclidean distance between views $i$ and $j$. The next training view $j$ is then sampled from the $k$-nearest neighbours of the current view $i$ according to the distribution: $\mathbb{P}(j \mid i) \propto \frac{1}{w_{ij}+W}$, where $W$ is a normalization constant that also controls the degree of exploration.

However, care must be taken when modifying the order of training views, as deviating from uniform random sampling may introduce training bias. To counteract this, we inject a randomly selected view every 128 iterations, which we find sufficient to preserve generalization performance.

### 4.5 MEMORY LAYOUT

Figure 6 illustrates the peak memory usage for a single training iteration of a 60-million-Gaussian hierarchy on the MC-smaller-city+ dataset. The output frame from this iteration is shown in Figure 12. The majority of RAM usage is consumed by per-Gaussian properties and their corresponding ADAM optimizer states. In contrast, the hierarchy structure itself accounts for less than 10% of the total RAM footprint. On the GPU, the SPT metadata for all 60 million Gaussians occupies just 680 MB of VRAM and the upper hierarchy negligible 24 MB.

Even in wide-angle aerial views, only a subset of the scene is actively loaded into GPU memory. In the example shown, 2.2 million Gaussians are rendered directly, while an additional 2.4 million are retained in the cache for future use. The bulk of GPU memory is instead consumed by temporary allocations for rasterization and optimization, which scale with the number of Gaussians rendered. Therefore, minimizing the number of active Gaussians is critical for staying within the VRAM budget.

The remaining GPU memory usage consists of auxiliary data, including cache management, hierarchy cut tracking, training and ground-truth images, as well as general PyTorch overhead. Figure 14

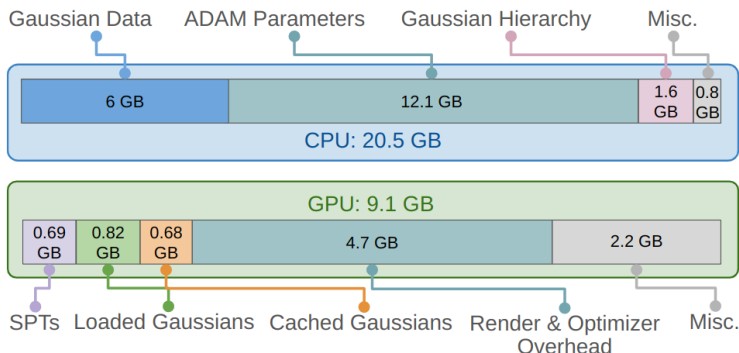

Figure 6: Peak memory consumption of CPU and GPU for a training iteration on MC-smaller-city+ with 60 million Gaussians.

provides a breakdown of all data associated with a single Gaussian and how they are distributed across CPU and GPU.

## 5 EVALUATION

Our method is designed to enable seamless training and rendering on ultra-large-scale scenes comprising tens of thousands of views captured at vastly different scales. Unfortunately, most datasets of sufficient size contain either street-level or aerial views—but not both. To overcome these challenges, we turn to the synthetic *MatrixCity* dataset (Li et al., 2023), which provides dense multi-scale views along with ground-truth camera poses and sparse geometry. We also include results on the real-world street-level datasets from H-3DGS. As baseline, we compare to recent large-scale 3DGS methods *CityGaussian* (Liu et al., 2024), H-3DGS (Kerbl et al., 2024), *HorizonGS* (Jiang et al., 2024) and *OctreeGS* (Ren et al., 2024).

With the exception of Kerbl et al. (2024), the biggest datasets these methods use in their papers consist of just 5620 images. In order to train MC-small-city+ with 42.2 thousand images, it was necessary to modify CityGaussian, HorizonGS and OctreeGS to load images from disk instead of caching them in RAM or VRAM. This slows down training, but does not affect the final results. In general, the hyperparameters suggested for large-scale scenes were used for baselines. If provided, we use configurations explicitly targeting the MatrixCity dataset, if it improves metrics over default configuration. Learning rates were not modified, as they were similar to ours and should not be a barrier to reconstruction on this scale. We also reached out to the authors of all our baselines about suggesting hyperparameters for the MC-small-city+ dataset, which we incorporated when provided. For HorizonGS, CityGaussian and our method, we include all used configuration files in the supplemental material. Additional details on hyperparameters and training for all compared methods can be found in appendix A.5. Further results and additional details on the datasets and baselines and are included in A.4 and method ablations can be found in A.3. Evaluations were run on a single H200 GPU with 141 GB VRAM, while all of our models were trained on a single RTX 3090 graphics card with 24 GB VRAM. We note that HorizonGS is not evaluated on the H-3DGS dataset, since it requires aerial views for training.

### 5.1 RESULTS

Qualitative comparisons are shown in Figure 7 and quantitative comparisons in Tables 1 and 2. The outputs of H-3DGS and *HorizonGS* show significant floating and ghosting artifacts. While H-3DGS performs well on individual chunks of MC-smaller-city+, oversized floaters survive the merging procedure and obscure the majority of test images (cf. Figure 11, 13 and 17). The merging procedure of *CityGaussian* (Liu et al., 2024), which is designed for aerial-only datasets, discards most of the trained Gaussians to avoid chunk artifacts, leading to mostly artifact-free views, but significantly blurry results. *A LoD of Gaussians* reconstructs novel views with a high degree of detail while avoiding chunk-based artifacts (cf. Figure 16), resulting in a considerable lead in quality

Table 1: **MatrixCity novel view synthesis results**. Results with * use the MC-smaller-city+ dataset due to memory constraints. Results with † require suboptimal COLMAP initialization instead of the dataset's provided point cloud. VRAM usage is measured while rendering the test images. Methods are considered out-of-memory (OOM) if they exceed 141GB of VRAM during rendering or training.

| Method | MC-small-city+ (42.2k images) / *MC-smaller-city+ (15.1k images) | | | | | |
| | PSNR$^\uparrow$ | SSIM$^\uparrow$ | LPIPS$^\downarrow$ | VRAM$^\downarrow$ | #iterations$^\downarrow$ | #Gaussians |
|---|---|---|---|---|---|---|
| CityGaussian | 19.78 | 0.650 | 0.475 | 2.4 GB | 1.11M | 2.7M |
| HorizonGS | 12.06 | 0.521 | 0.544 | 20.6 GB | 2.5M | 50.3M |
| H-3DGS | OOM | OOM | OOM | >141GB | OOM | OOM |
| OctreeGS | OOM | OOM | OOM | >141GB | OOM | OOM |
| Ours | 21.59 | 0.711 | 0.396 | 25.0GB | 600k | 136.7M |
| HorizonGS* | 15.28 | 0.601 | 0.435 | 13.91 | 1.2M | 34.1M |
| OctreeGS* | OOM | OOM | OOM | >141GB | OOM | OOM |
| Ours* | 21.71 | 0.710 | 0.370 | 17.7GB | 310k | 60M |
| H-3DGS*† | 14.57 | 0.516 | 0.563 | 97GB | 4M | 83.1M |
| Ours*† | 20.59 | 0.658 | 0.468 | 21.7GB | 310k | 24.7M |

metrics. Seamless training enables faster convergence during training, reducing the required number of training iterations considerably over divide-and-conquer based methods.

Table 2: **Hierarchical 3DGS dataset novel view synthesis results.**

| Method | Small City (5.8k images) | | | Campus (22.0k images) | | |
| | PSNR$^\uparrow$ | SSIM$^\uparrow$ | LPIPS$^\downarrow$ | PSNR$^\uparrow$ | SSIM$^\uparrow$ | LPIPS$^\downarrow$ |
|---|---|---|---|---|---|---|
| OctreeGS | 19.20 | 0.601 | 0.498 | OOM | OOM | OOM |
| CityGS | 21.40 | 0.692 | 0.383 | 19.47 | 0.645 | 0.531 |
| H-3DGS (leaves) | 24.35 | 0.788 | 0.273 | 17.84 | 0.6101 | 0.4667 |
| Ours | 24.61 | 0.774 | 0.297 | 21.85 | 0.707 | 0.456 |

**Rendering** Our level-of-detail and caching strategy can also be applied to efficiently render the finished models: As supplemental material, we include fly-through videos of all scenes rendered on an RTX 3090 GPU at this anonymized link: `https://www.youtube.com/playlist?list=PLocO6QqSKZ4YN7DyuC5gWEeVciSV6RdAk`. Table 1 demonstrates effective VRAM reduction during rendering of test images in comparison with other large-scale 3DGS methods. Comparisons of rendering VRAM with 3DGS (Kerbl et al., 2023) and gsplat (Ye et al., 2025) are included in Appendix A.4.1.

# 6 DISCUSSION AND OUTLOOK

*A LoD of Gaussians* enables seamless training and rendering of ultra-large 3DGS models on consumer hardware. By storing Gaussian data in external memory and streaming it on demand, our method avoids the pitfalls of chunk-based pipelines. The hierarchical SPT structure accelerates LoD selection and remains robust to ongoing training changes. Combined with caching and view selection, our system significantly reduces out-of-core overhead. Together, these components enable efficient reconstruction and rendering at scale, demonstrated on the MC-small-city+ dataset.

**Limitations and Future Work** A primary limitation of our method lies in initialization. Accurate estimation of camera poses and sparse point clouds remains challenging at large scale, particularly for real-world datasets with sparse or inconsistent coverage. Future work could explore improved initialization strategies or joint optimization of poses and Gaussians during early training.

Our system also requires roughly 1 GB of RAM per million Gaussians, which—while more efficient than prior methods—still constrains scalability. Loading from disk is feasible in our experiments, though at the cost of a $10\times$ slowdown. Further, while our method in general greatly reduces the number of training iterations necessary, individual training iterations can take a lot longer than standard 3DGS due to the loading overhead.

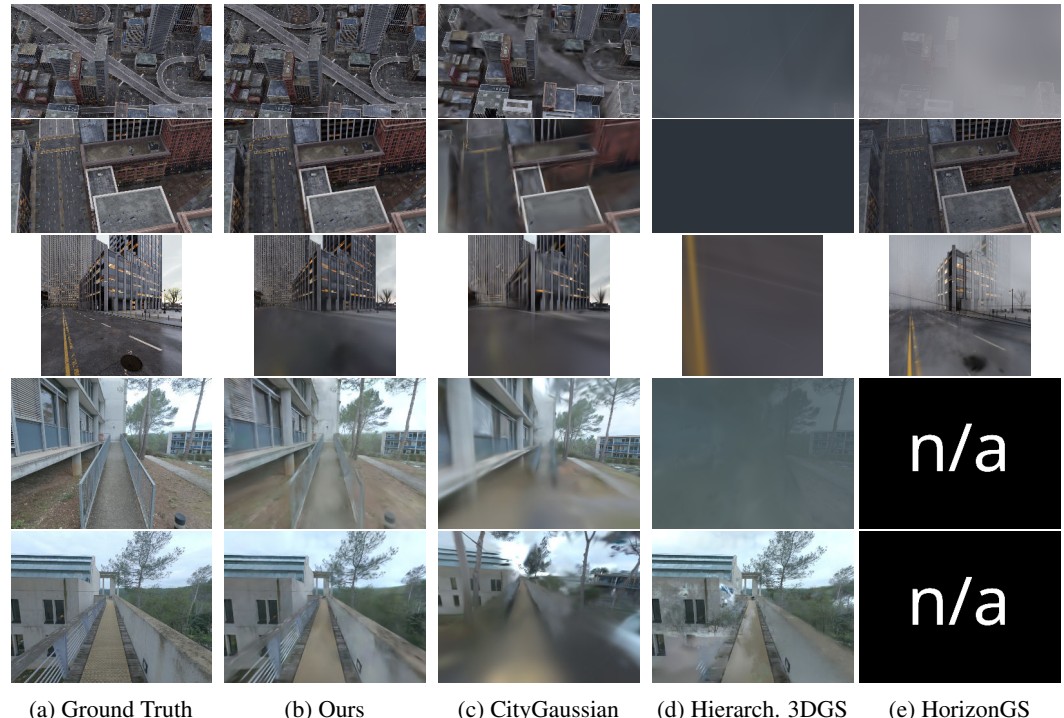

|  (a) Ground Truth | (b) Ours | (c) CityGaussian | (d) Hierarch. 3DGS | (e) HorizonGS |

Figure 7: Qualitative comparison of our method and SOTA methods (Liu et al., 2024; Kerbl et al., 2024; Ren et al., 2024) on the MC-small-city+ and campus datasets.

The level-of-detail system makes our method robust against vast differences in view distance, but when these differences are not present, like in single-height aerial datasets as evaluated in Appendix A.4, it becomes a burden instead, making more straight-forward divide-and-conquer training the better choice.

Interactive rendering could be further optimized by avoiding per-frame hierarchy cut recomputation and enabling asynchronous streaming. While frustum culling reduces memory load in most views, it is ineffective when the entire scene falls inside the frustum. Occlusion culling could address this by skipping entire SPTs before loading.

Overall, we believe that out-of-core 3D Gaussian Splatting is a promising direction for scaling radiance field methods to city-scale and beyond, without the need for specialized hardware.

**Reproducibility Statement**    To ensure reproducibility of our results, we have appended our entire codebase along with the config files of our method and baselines required to train each of the scenes. Detailed breakdowns of datasets and hyperparameters can be found in Appendix A.4 and A.5.

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

# A APPENDIX

## A.1 FIGURES

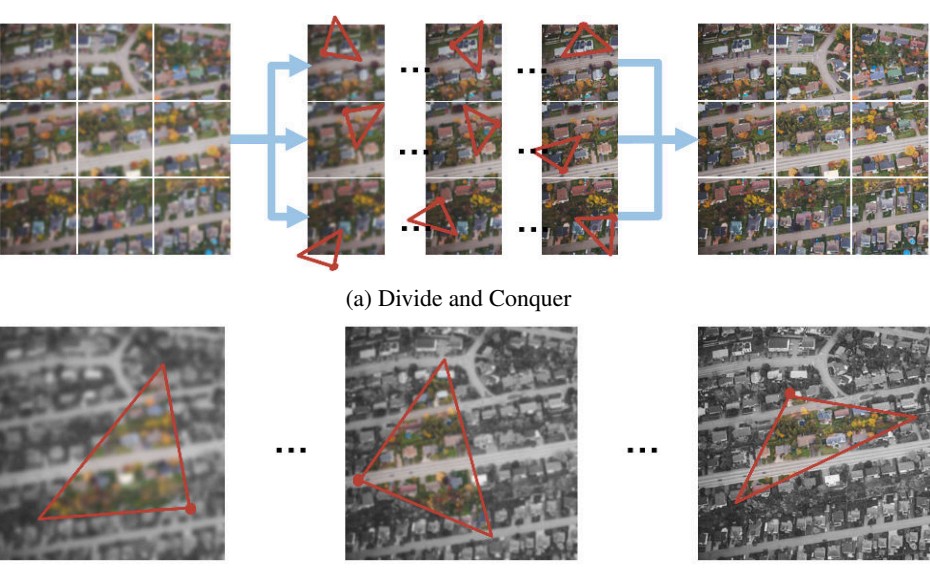

(a) Divide and Conquer

(b) Ours

Figure 8: Comparison between the divide and conquer training process as used in Kerbl et al. (2024); Lin et al. (2024); Liu et al. (2024); Jiang et al. (2024) to our training process. Colored regions are present in VRAM, training views are drawn in red.

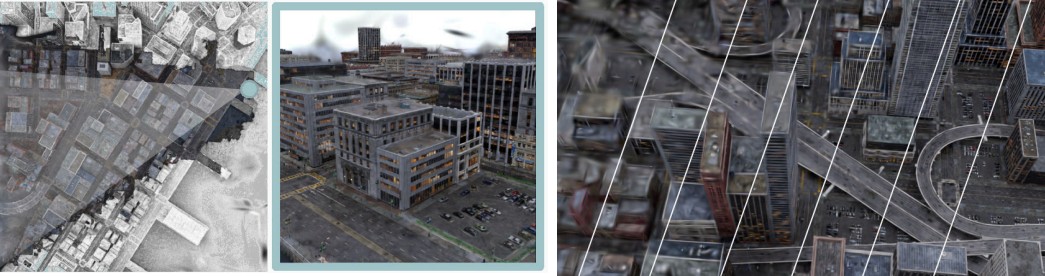

Figure 9: Frustum Culling and LoD selection (left) greatly reduces the number of Gaussians required to render a view (right).

Figure 10: Hierarchical SPTs enable smooth transitions between detailed (left) and coarse (right) representation.

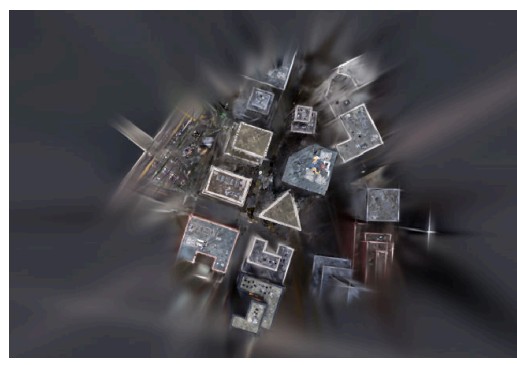 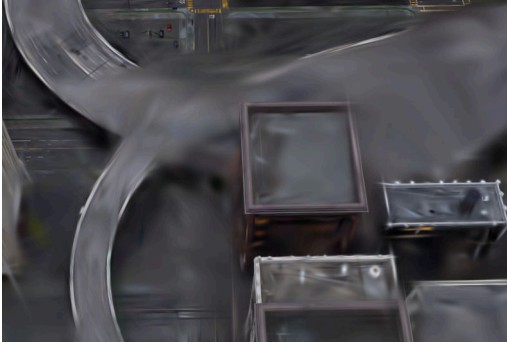

(a) Chunk Bleeding in H-3DGS (Kerbl et al., 2024).  (b) Chunk Ghosting in CityGaussian (Liu et al., 2024).

Figure 11: Artifacts caused by the divide-and-conquer strategy on the MC-small-city+ dataset.

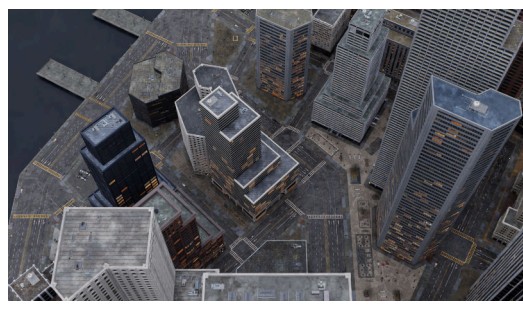 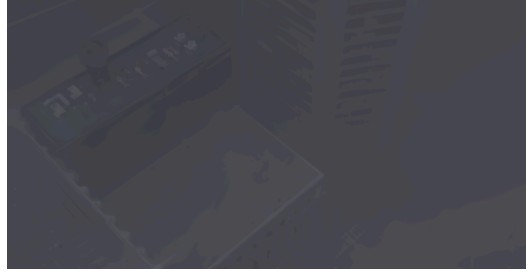

Figure 12: Training image rendered during the iteration depicted in Figure 6.

Figure 13: Images from H-3DGS on the MatrixCity-Scale scene are obscured by large floaters.

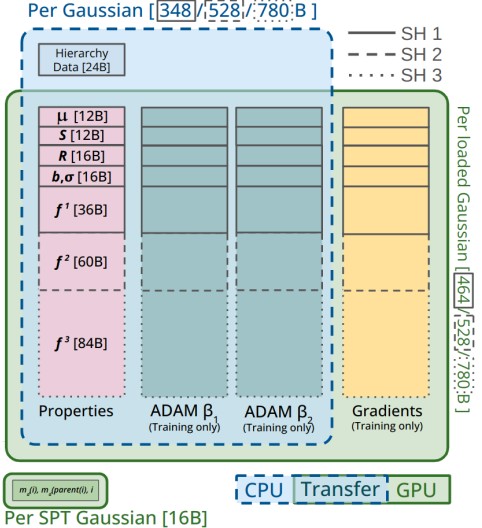 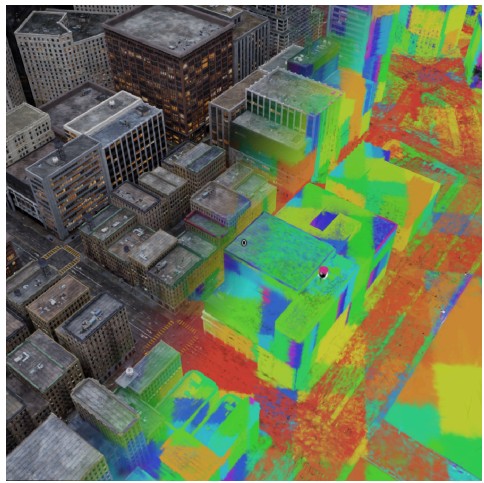

Figure 14: All Gaussian properties except gradients are always present in CPU RAM, whereas only the currently loaded Gaussians and slim SPT information needs to be stored on GPU.

Figure 15: SPTs for a frame of MatrixCity rendered in different colors.

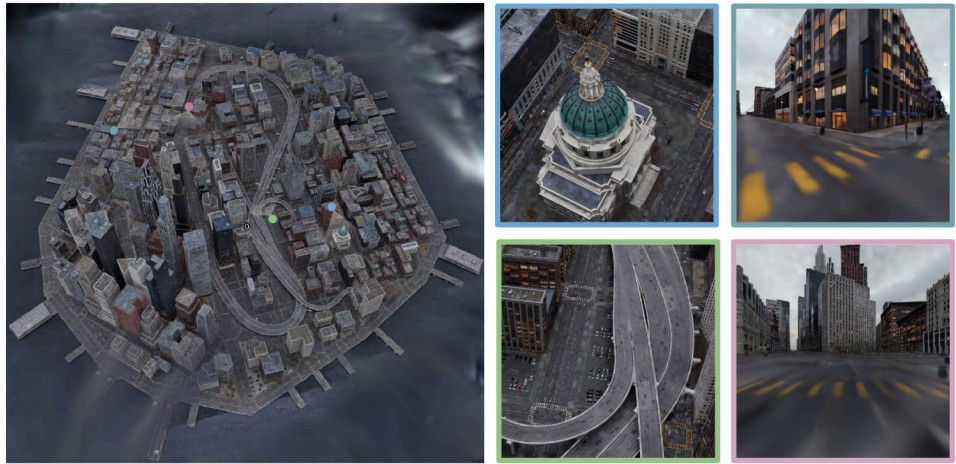

Figure 16: *A LoD of Gaussians* is able to reconstruct and render huge environments with 150M+ Gaussians with large variations of scale on consumer hardware.

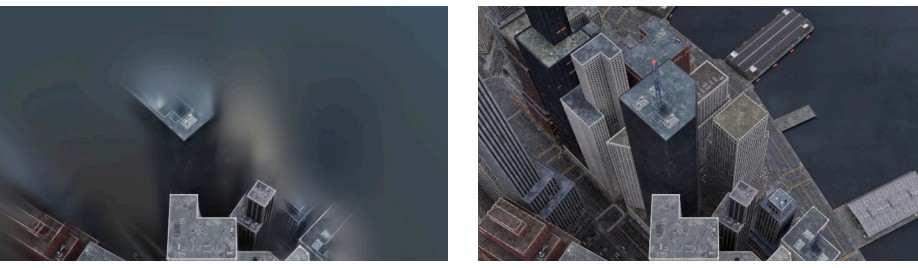

(a) Image generated during chunk training by *H-3DGS*.

(b) The same image generated during training by *A LoD of Gaussians*.

Figure 17: Chunk-based optimizations causes floaters to expand outside of the chunk boundary.

## A.2 INITIALIZATION AND TRAINING

Following Kerbl et al. (2024), we initialize the Gaussian model from a sparse point cloud, augmented with skybox points. This initial representation is small enough to fully reside in GPU memory and is trained for 100k iterations without densification. The goal of this phase is to establish a stable global scene structure before constructing the hierarchy. After this initial optimization, we build a binary Gaussian hierarchy, where the trained Gaussians act as leaf nodes and the parent nodes represent merged approximations of their two children.

The Gaussian properties–including base color, SH coefficients, position, and covariance–are optimized using the standard loss propagation introduced by Kerbl et al. (2023). To fulfill our tight memory requirements, we update only the parameters of the cut set chosen for each training view. Thus, gradients may, in general, be propagated to Gaussians in the middle of the hierarchy. Assuming good coverage and diversity of training views, these updates will diffuse over the entire hierarchy over the course of training, leading to smooth transitions from the highest to the lowest LoD (see Figure 10).

## A.3 ABLATIONS

We assess the contribution of key components through an ablation study, using recorded camera paths across all scenes (see supplemental videos). Table 3 reports average frame times over these paths. Caching significantly improves rendering performance, roughly doubling the framerate across scenes by reducing the average number of Gaussians that need to be loaded from RAM by 93% on campus and 86% on MC-small-city+. The effectiveness of frustum culling scales with scene size: On the largest MC-small-city+ scene, 24.5 million Gaussians are frustum culled on average (a reduction of 88%), while for campus and small-city, from H-3DGS the average number was 9.8 million and and 7.9 million respectively (a reduction of 74% and 65%). Thus, for small datasets like campus (38M) the computational cost of frustum cullling exceeds the performance gains, but it is essential for large scenes.

To evaluate cut efficiency, we compare the time required to compute the visible set using either full hierarchy BFS or our HSPT-based approach. HSPT consistently yields faster cut times, which can be attributed to better parallelization. Further, the BFS approach requires positions and scales for all Gaussians to be present in memory, causing it to exceed 24GB of VRAM on MC-small-city+, while the HSPT method peaks at 21GB. For ablating the training process, we measure average iteration durations over 1 000 steps. Here, frustum culling and caching prove essential, substantially reducing the number of Gaussians loaded and rendered per view.

Table 3: **Ablations.** Average frame times for rendering camera paths and average iteration times during training with and without caching Gaussians. The final results show the average timings of the hierarchy cut during rendering using our HSPT and the baseline BFS approach. For campus, we evaluate two differently sized models with 38M and 80M Gaussians respectively.

|  | MC-smaller-city+ 60M | MC-small-city+ 136M | Campus 38M | Campus 80M |
|---|---|---|---|---|
| Render Time Full | 48.1 ms | 68.3ms | 47.1 ms | 83.2 ms |
| Render Time w/o Cache | 119.4 ms | 127.4 ms | 92.6 ms | 222.3ms |
| Render Time w/o Frustum Culling | 52.3 ms | 89.3 | 38.3 ms | 110.2 ms |
| Train Time Full | 156 ms | 174 ms | 205 ms | |
| Train Time w/o Cache | 471 ms | 257 ms | 244 ms | |
| Train Time w/o Frustum Culling | 685 ms | 626 ms | 312 ms | |
| Render Cut Time (HSPT) | 31.9 ms | 45.7ms | 31.3 ms | 36.5 ms |
| Render Cut Time (BFS) | 47.8 ms | OOM | 40.0 ms | 53.7 ms |

**View Selection** When training MC-smaller-city+ dataset (250k iterations, up to 60M Gaussians), enabling our guided view selection reduces the number of Gaussians loaded from RAM per frame from 355 770 to 219 043—a 35% reduction in memory transfers. Reconstruction quality is unaffected, with PSNR and SSIM improving slightly by 0.04 and 0.05, respectively.

**Effectiveness of LOD**  To evaluate the effectiveness of our level-of-detail system, we choose 5 random images from each dataset and compare the resulting quality metrics to the number of Gaussians rendered at 50 different levels of details. The results are visualized in Figure 18 in addition to average FPS for rendering the view 10 times with pre-warmed cache on an H200 GPU. As expected, FPS scales closely with the inverse of the number of Gaussians. In the vast majority of cases, increasing the LOD level consistently improves rendering quality. Overall, we only notice a small dip in quality when reducing the Gaussian count by 50%, verifying the effectiveness of our LoD structure. Further, using continuous instead of discrete levels of detail leads to the smoothness of the curves in Figure 18. This also indicates a smooth transition between LoD levels, which reduces popping.

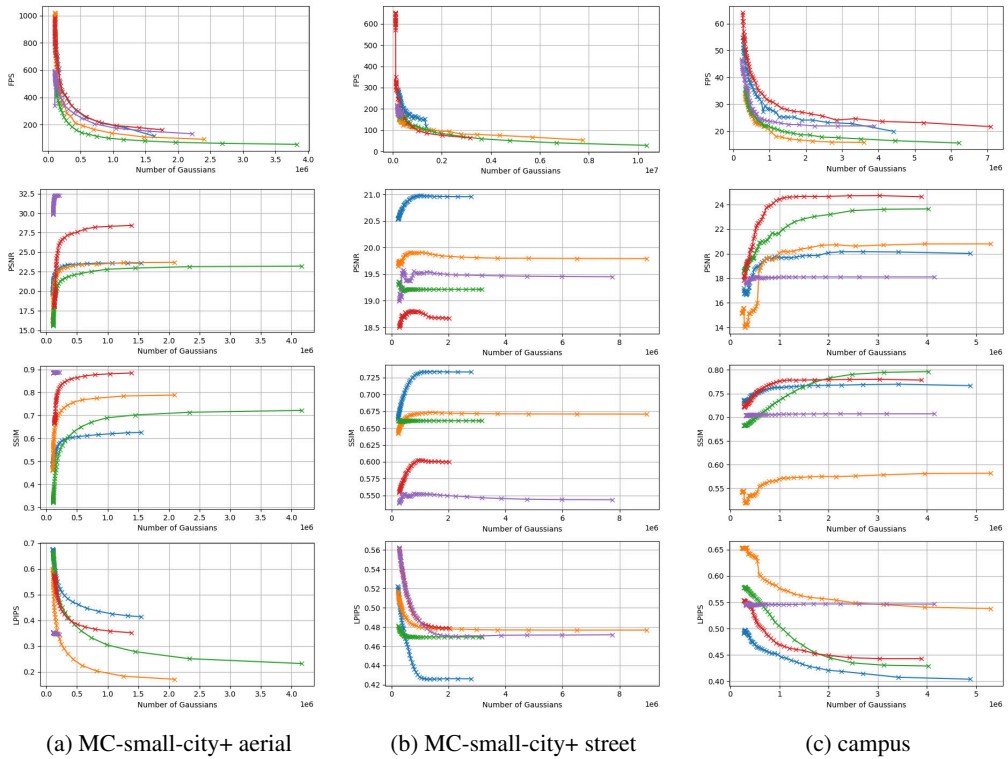

      (a) MC-small-city+ aerial        (b) MC-small-city+ street        (c) campus

Figure 18: Qualitative evaluation of our LoD system for 5 randomly chosen views and 50 different levels of detail.

**Effectiveness of Caching**  In Figure 19, we provide sensitivity curves for cache size vs. FPS during rendering. In each iteration up to *cache size* Gaussians from the previous frames may be reused. The results once again demonstrate the importance of the caching system to rendering efficiency. The graphs show that performance increases steeply with increasing cache size until it exceeds the typical number of Gaussians required per iteration and starts to plateau.

## A.4 DATASET DETAILS AND FURTHER RESULTS

We include more details on the datasets used in the evaluation and provide additional results for smaller-scale scenes.

**MC-small-city+**  For our main benchmark, *MC-small-city+*, we aggregate 33 006 street-view images and 7 672 aerial views from the small-city scene of the MatrixCity dataset (Li et al., 2023) and add 533 additional high-altitude views we generated manually. We evaluate reconstruction quality on a separate set of 4 228 test views, according to the test split provided by the dataset. The scene is extremely challenging due to its enormous scale, sparse views and wide variation in scale: As such, we use a subset of the entire dataset (15.1k images, covering about a third of the area), denoted as

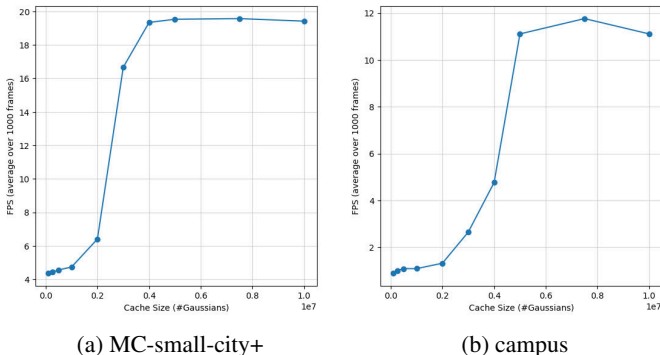

(a) MC-small-city+                (b) campus

Figure 19: Average FPS for rendering 1000 frames of the camera paths for the *MC-small-city+* and *campus* scenes with various cache sizes.

MC-smaller-city+ for baselines that are not able to reconstruct the full dataset. For MC-small-city+, we use the camera poses provided by the MatrixCity dataset and convert them to the COLMAP format for Gaussian splatting. We merge the provided street and aerial sparse point clouds and randomly downsample them by a factor of 5 to get more realistic initialization conditions. We do not make use of the ground truth depth images provided by the dataset, as we consider this to be an unrealistic advantage. Because the dataset does not provide point correspondences required for scaling monocular depth maps, we deactivate depth supervision in our method and baselines for this dataset. Figure 20 shows the scene along with the camera distribution. While some methods have successfully reconstructed only aerial (Liu et al., 2024; 2025; Ren et al., 2024) or only street-level views (Liu et al., 2025; Li et al., 2024b), training a model that holds up to scrutiny from both perspectives presents a particular challenge. Training on close and far views simultaneously–without a proper LoD system like ours in place–significantly degrades visual quality for both, as noted by Jiang et al. (2024); Zhang et al. (2024). Moreover, such a scenario complicates partitioning the scene into independent chunks.

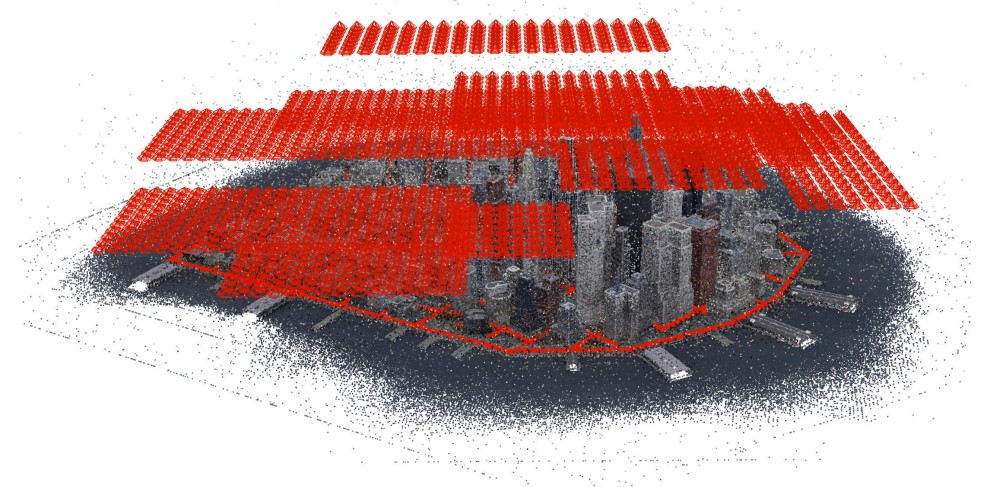

Figure 20: The MC-small-city+ dataset spans multiple city blocks, which are supervised by tens of thousands of views, varying widely in scale.

**Hierarchical 3DGS dataset**   H-3DGS Kerbl et al. (2024) did not use a test split for their dataset, but we hold back every 100th image alphabetically for testing. We reevaluate H-3DGS for this scene with the new test split, using the camera calibrations and chunk splits provided on their website. In accordance with the instructions, we disable exposure optimization for evaluation. On this dataset,

Table 4: **H-3DGS single chunk view synthesis results.** Results reproduced from Kerbl et al. (2024).

| Method | Small City Chunk (1.1k images) | | | Campus Chunk (1.5k images) | | |
|---|---|---|---|---|---|---|
| | PSNR$\uparrow$ | SSIM$\uparrow$ | LPIPS$\downarrow$ | PSNR$\uparrow$ | SSIM$\uparrow$ | LPIPS$\downarrow$ |
| 3DGS | 25.34 | 0.776 | 0.337 | 23.87 | 0.785 | 0.378 |
| H-3DGS (leaves) | 26.62 | 0.820 | 0.259 | 24.61 | 0.807 | 0.331 |
| Ours | 25.94 | 0.808 | 0.276 | 24.86 | 0.796 | 0.360 |

we use depth supervision identical to Kerbl et al. (2024).

**Hierarchical 3DGS single-chunk dataset**    To demonstrate our ability to reconstruct small scenes, we also evaluate our method on the smaller, single-chunk versions of the *Campus* and *Small City* (cf. Table 4). The quality metrics indicate that training on the LoD structure (which would not be required for scenes of this scales) during training only leads to marginal reductions in visual quality. The distribution of street views in the campus dataset is visualized in Figure 21b.

**Mill19 dataset**    Table 5 shows results on the Mill19 dataset (Turki et al., 2022). We use the camera poses and sparse point cloud provided by Liu et al. (2024). In accordance with baselines, we downscale all images by a factor of 4. We have chosen this dataset, because it is widely used in large-scale novel-view synthesis. At the same time, this dataset represents a worst-case scenario for our method, as the regular, same-height aerial views (cf. Figure 21a) negate any benefit of our level-of-detail method and can be split into independent chunks trivially.

Table 5: **Mill19 novel view synthesis results** Results with $\dagger$ are reproduced from Liu et al. (2024). Results of H-3DGS are reproduced from Kerbl et al. (2024).

| Method | Rubble (1.6k images) | | | | Building (1.9k images) | | | |
|---|---|---|---|---|---|---|---|---|
| | PSNR$\uparrow$ | SSIM$\uparrow$ | LPIPS$\downarrow$ | Size | PSNR$\uparrow$ | SSIM$\uparrow$ | LPIPS$\downarrow$ | Size |
| MegaNeRF[†] | 24.06 | 0.553 | 0.516 | n/a | 20.93 | 0.547 | 0.504 | n/a |
| 3DGS[†] | 25.47 | 0.777 | 0.277 | 6.1M | 20.46 | 0.720 | 0.305 | 6.4M |
| CityGaussian[†] | 25.77 | 0.813 | 0.228 | 9.7M | 21.55 | 0.778 | 0.246 | 13.2M |
| H-3DGS | 24.64 | 0.755 | 0.284 | n/a | 21.52 | 0.723 | 0.297 | n/a |
| Ours | 23.75 | 0.704 | 0.330 | 25M | 20.67 | 0.682 | 0.318 | 25M |

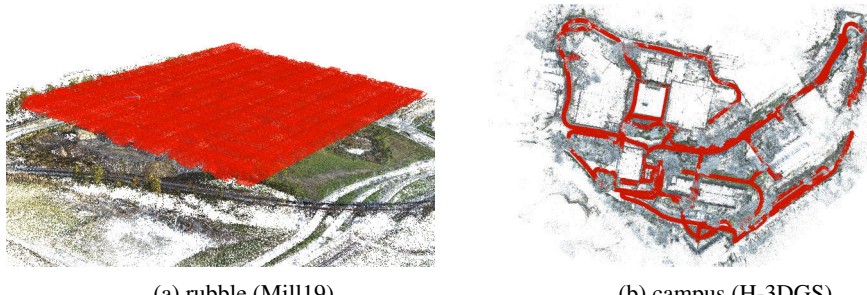

(a) rubble (Mill19)                                   (b) campus (H-3DGS)

Figure 21: Overview of the aerial and street datasets.

### A.4.1 RENDERING COMPARISON

To demonstrate the effectiveness of our LoD method, we compare VRAM usage and image quality to rendering the highest level of detail leaf Gaussians with both 3DGS (Kerbl et al., 2023) and gsplat (Ye et al., 2025) using their default rasterization settings. We conducted comparisons across three different viewpoints to assess performance at varying levels of detail. As illustrated in Figure 22, our approach achieves visual fidelity comparable to the baselines while significantly reducing VRAM usage, highlighting the effectiveness of our LOD scheme.

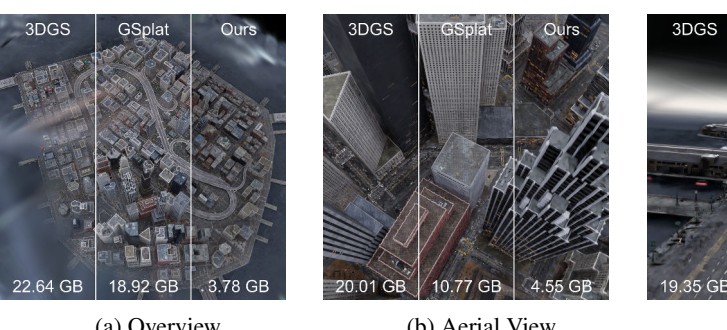

| (a) Overview | (b) Aerial View | (c) Street View |

Figure 22: Comparision to standard 3DGS (Kerbl et al., 2023) and GSplat (Ye et al., 2025) in terms of rendering quality and VRAM usage.

### A.5 EVALUATION DETAILS

Finding baselines for the MC-small-city+ scene proved challenging, as the only methods we are aware of that have successfully trained on all fused aerial and street-level views–Li et al. (2024a) and Li et al. (2024b)–both require 64 GPUs running in parallel. We were not able to meet these hardware requirements for evaluation and hold that these methods and similar multi-GPU works like Zhao et al. (2024) are orthogonal to our method. H-3DGS (Kerbl et al., 2024) has demonstrated the ability to reconstruct similar sized datasets, but its evaluation was restricted to street-level views. We choose Kerbl et al. (2024) and other popular large-scale Gaussian Splatting methods (Liu et al., 2024; Jiang et al., 2024; Ren et al., 2024) as baselines. Most of these large-scale 3DGS frameworks have complicated multi-stage training processes and are very sensitive to hyperparameters, which we detail in the following section. We consider it a benefit of our training pipeline that the user is not confronted with tuning the error-prone partitioning process, being more akin to the original Gaussian Splatting training.

**OctreeGS** For *OctreeGS* (Ren et al., 2024), we follow the provided instructions on training custom datasets. Note that OctreeGS does not perform any scene division and uses the memory-intensive ScaffoldGS (Lu et al., 2023) method, making it unsuitable for scenes of this scale.

**CityGaussian** We choose *CityGaussian* (Liu et al., 2024) (specifically the 1.2 version of the repository) as a baseline instead of *CityGaussianV2* (Liu et al., 2025), as the code release for *CityGaussianV2* does not yet support level-of-detail rendering. We follow the instructions on training large datasets and use the parameters and chunk split from the provided configuration file for the MatrixCity-Aerial dataset (which covers the same region as the MC-small-city+ dataset, but without the street views), as they produced better metrics than the default parameters. Note that the low number of Gaussians in the final model is due to *CityGS* discarding most (about 90%) of the trained Gaussians after chunk training in order to avoid artifacts. For campus and small city, we used a $4 \times 4$ and $2 \times 2$ chunk split to match the number of chunks as closely as possible with the 12/4 chunks used by H-3DGS (Kerbl et al., 2024).

**Hierarchical-3DGS** We follow the instructions of Kerbl et al. (2024) for running the method on large scenes. Note that we do not make use of their particular COLMAP pipeline for MC-smaller-city+, as it only works with unbroken camera paths. We disable exposure optimization for evaluation in accordance with the instructions. The partitioning step of H-3DGS requires point correspondences

not provided by the dataset. To substitute, we use the camera poses provided by the dataset and generate the sparse point cloud and correspondences from scratch using COLMAP. On scenes of this scale, COLMAP output contains significant noise, which also leads to reduced quality metrics for our method.

H-3DGS achieves drastically worse PSNR and SSIM results on the full campus dataset, compared to the single-chunk results, due to a slight perspective distortion that occurs specifically on this dataset (it did not occur on small city). This can be reproduced by evaluating the trained model on their campus dataset, both of which are available on their website. As such, we include the results, but encourage visual comparison in Figure 7.

**HorizonGS**   We expand the configuration provided for a single chunk of the MatrixCity small-scene dataset (named *ours/large_scene/block_A*) with a chunk split of $3 \times 4$ and $5 \times 5$ for MC-smaller-city+ and MC-small-city+ (shown in Figure 23) respectively. The partitions of the MC-small-city+ scene Each chunk is trained for 60000 iterations and then fine-tuned for another 40000.

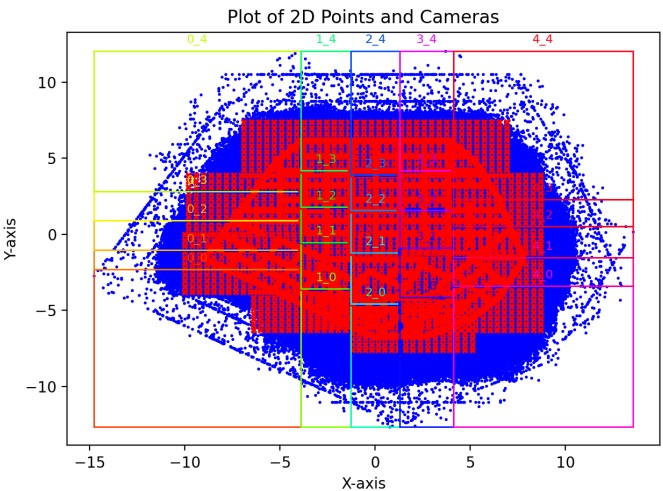

Figure 23: Partitions of HorizonGS on the MC-small-city+ scene.

**VastGaussian**   *VastGaussian* (Lin et al., 2024) would have presented a natural comparison point, but there is currently no official code release available.

**Ours**   We provide detailed configuration files for all of the scenes as supplemental material. In general, we perform 60k iterations of coarse training (except for 100k on MC-small-city+) and then 150k iterations of fine training (except 250k on campus and MC-smaller-city+ and 500k on MC-small-city+). We shoose SH degree 1 and half the densifaction gradient threshold as opposed to H-3DGS. We pick a cache size of 15000000 Gaussians.

A.6   IMPLEMENTATION DETAILS

Training models on this unprecedented scale on consumer hardware required solving many technical issues. In this section, we elaborate on the most important of these design decisions.

**Codebase**   Our general code structure is based on H-3DGS (Kerbl et al., 2024). In particular, we reuse the hierarchy creator, but replace the cut procedure and chunk based training, as well as the rasterizer for the one used in Kerbl et al. (2023). We also use the densification implementation of Kheradmand et al. (2024). All of the code is implemented in PyTorch and C++/CUDA. The entire source code is included as supplemental material.

**Varying level of detail**   We introduce noise into the hierarchy cuts in order to prevent overfitting of Gaussians of a certain scale to views from a particular distance. In particular, we multiply the

distances to the SPT centers each iteration with $1 + 5r^4$, with a uniformly random variable $r \in U(0, 1)$. This factor is designed such that most iterations will train near the highest level of detail, but coarser levels will also be trained occasionally. While this does not improve image metrics on the test set, it shows significant improvement to out-of-distribution views and reduces LoD popping artifacts.

**Varying focal length**    The required level of detail is not only dependent on a camera's distance $d$ to the Gaussian, but also on its focal length, which we need to account for as our datasets contain views with differing focal lengths. Therefore, we choose a base focal length $f_b$ and use a relative distance metric $\hat{d}$ for our HSPT cuts, which is calculated for the current camera's focal length $f_i$ as $\hat{d} = \frac{f_b}{f_i} d$. E.g. for a camera with double the focal length, the cut distance should be halved to account for the larger projection footprint on the image plane.

**SH Degree**    For the most part, experiments indicate that SH degrees higher than 1 do not significantly contribute to image quality in the tested scenes. For training higher degrees $n$ of SH, we have found that increasing the degree from 0 to 1 during the course stage and then gradually increasing the degree from 1 to $n$ during fine training (with each increase happening after 10% of total training iterations) yields the best results.

**Gaussian Order**    To exploit spatial coherency and improve training performance, we store the Gaussians on CPU in Morton Z-order. As this order can change during training, we resort the Gaussians at every densification iteration.

**Training Images**    Conventional Gaussian Splatting stores the entire training dataset in VRAM to achieve their impressive training speed, however, this approach is infeasible for training ultra-large-scale datasets on consumer-grade hardware. Instead, we load the ground truth images from disk every iteration, saving VRAM at the cost of training performance.

**Unreachable Gaussians**    During training, it can occur that a child Gaussian becomes larger than its parent. This can lead to cases where $m_d(\text{parent}(i)) < m_d(i)$, making it impossible for the parent Gaussian to fulfil the SPT cut condition. While we are aware of this inefficiency, we found that in practice this only happens to a small number of Gaussians ($< 10\%$). They still occupy a portion of RAM, but are never rendered or transferred. We have experimented with rebalancing the hierarchy during training, but found it too costly for little benefit. In general, we find this behaviour to be preferable over generating improper cuts, which can derail training.

**Respawning, Densifying, and Pruning Gaussians**    For improved performance, all respawn- and densify-operations are performed in parallel during the densification step. This can lead to difficult edge cases that need to be handled to prevent the hierarchy from degenerating: When two sibling nodes both need to be respawned, we only respawn the right node. This will cause the left node to become a new dead leaf, which will be respawned in the next densification iteration. If a node needs to be respawned whose sibling is not a leaf node, its entire subtree will replace the parent node.
To minimize the number of unnecessary Gaussians, we apply a simple pruning strategy where we zero the opacity of Gaussians that have not contributed for a number of iterations equal to twice the size of the training set. This will cause them to be respawned in the next densification iteration.

**Performance Optimization**    The performance of the HSPT cut and Gaussian loading is particularly important, as they happen every iteration. We store the SPT properties ($m_d(i)$, $m_d(\text{parent}(i))$, $i$) for all Gaussians in a single continuous GPU memory buffer and perform the cuts in parallel using optimized CUDA kernels. Similarly, the Gaussian properties are stored in a single PyTorch tensor in RAM as an array of structures (each Gaussian's properties concurrently). The properties of all required Gaussians are then transferred to the GPU via a single copy operation, reorganized on the GPU in a structure of arrays layout (as required by the rasterizer), and appended to the current render set.

## A.7    ADDITIONAL DENSIFICATION DETAILS

Our densification method combines the split operation from Kheradmand et al. (2024) with the selection method from Kerbl et al. (2024). Relying purely on the densification strategy of Kheradmand

et al. (2024)–applying a noise to every Gaussian's position and an opacity/scaling loss–becomes problematic for large scenes, where a majority of Gaussians are not visible from any one view: The result is a gradual disappearance of density in the scene or extreme "stringing" artifacts (cf. Figure 24). However, without these losses to encourage respawning of Gaussians, the densification becomes aimless and random. We implemented another strategy that only applies the losses and noise to Gaussians that affected at least one pixel in the current iteration. While this significantly improved results, we found that the MCMC densification strategy in general performs poorly when Gaussian density is low, which is necessitated by ultra-large-scale scenes. On small scenes such as the single-chunk datasets, our modified MCMC strategy slightly outperforms the presented strategy given sufficient Gaussian budget, but it lags behind in the larger scenes.

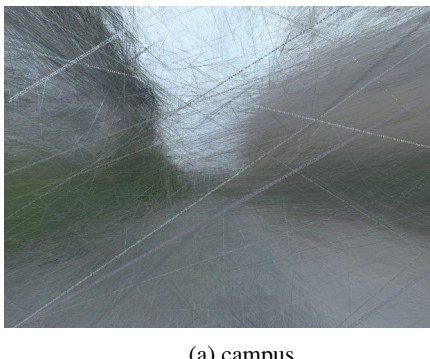 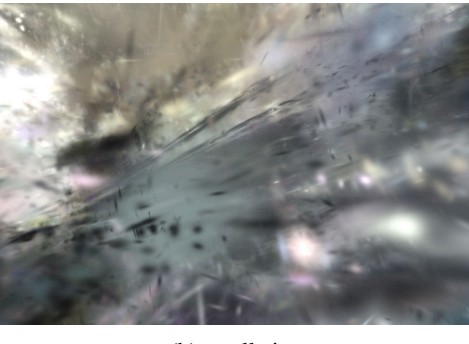

(a) campus                               (b) small city

Figure 24: Without regularization, the scale- and opacity-losses of 3DGS-as-MCMC lead to either stringing or disappearance of the entire scene.

## A.8 Pseudocode

This section provides pseudocode for our core algorithms. Algorithm 1 contains a procedure to "cut" a binary tree hierarchy along a certain condition. Algorithm 2 shows how to cut a single SPT at a particular distance $d^j$. The entire training procedure is sketched in Algorithm 3.

---

**Algorithm 1** BFS Hierarchy Cut

---

**procedure** CUT(Hierarchy $\mathcal{H}$, $condition$)
    $cut \leftarrow \{\}$
    $Q \leftarrow$ QUEUE($root(\mathcal{H})$)
    **while** $Q$ not empty **do**
        $node \leftarrow Q$.DEQUEUE()
        **if** $condition(node)$ **then**
            $cut \leftarrow cut \cup node$
        **else**
            $Q$.ENQUEUE($node \rightarrow left$)
            $Q$.ENQUEUE($node \rightarrow right$)
        **end if**
    **end while**
    return $cut$
**end procedure**

---

## A.9 Supplemental Material

As supplemental material, we include the codebase used for this project, along with the configuration files for all of our scenes. We also include configuration files for our baselines, if applicable. Further, we provide videos at the following anonymous link: `https://www.youtube.com/playlist?list=PLocO6QqSKZ4YN7DyuC5gWEeVciSV6RdAk`. This includes fly-throughs of the evaluated scenes and a method overview video.

**Algorithm 2** SPT Cut

---

**procedure** CUT(SPT $\mathcal{S}$, distance $d^j$)
    $high \leftarrow$ BINARYSEARCH($\mathcal{S}_{max}, d^j$)
    $cut \leftarrow \{i \in [0, high] | \mathcal{S}_{min}^i < d^j\}$
    return $cut$
**end procedure**

---

**Algorithm 3** Full train procedure

---

**procedure** TRAIN(upper Hierarchy $\mathcal{U}, cache, skybox$)
    $view \leftarrow 0$
    $Cache\_Distances \leftarrow \{\}$
    $Cache\_SPTs \leftarrow \{\}$
    **while** $True$ **do**
        $view \leftarrow$ SAMPLE($knn\_graph, view$)          $\triangleright$ Find a new nearby view
        $cut\_condition \leftarrow [\lambda(i) = \neg\, is\_in\_frustum(\mu_i) \vee m_d(i) < T]$
        $upper\_cut \leftarrow$ CUT($\mathcal{U}, cut\_condition$)
        $SPT\_roots \leftarrow \{u \in upper\_cut | u$ contains an SPT$\}$
        $upper\_nodes\_to\_render \leftarrow upper\_cut \setminus SPT\_roots$
        $SPT\_distances \leftarrow \{\|\mu_r - p_{view}\|_2^2$ for $r \in SPT\_roots\}$
        $Cache\_Reuse\_SPTs \leftarrow \{s_i \in (Cache\_SPTs \cap SPT\_roots) | D_{min} \leq \left\|\frac{\mu_i - \mathbf{p}_{cam}}{Cache\_Distances_i}\right\|_2^2 \leq D_{max}\}$
        $load\_SPTs \leftarrow (SPT\_roots \setminus Cache\_Reuse\_SPTs)$
        $load\_Gaussian\_indices \leftarrow$ CUT\_SPTs($load\_SPTs, SPT\_distances[load\_SPTs]$)
        $load\_Gaussians \leftarrow$ LOAD($load\_Gaussian\_indices$)
        $Gaussians \leftarrow skybox \cup upper\_nodes\_to\_render \cup load\_Gaussians$
        $Gaussians \leftarrow Gaussians \cup$ LOAD\_CACHE($Cache\_Reuse\_SPTs$)
        RENDER\_AND\_OPTIMIZE($Gaussians$)
        $Cache\_SPTs \leftarrow Cache\_SPTs \cup load\_SPTs$
        **if** $len(Cache\_Gaussians) > max\_cache\_size$ **then**
            WRITE\_BACK\_LRU($Cache\_Gaussians$)          $\triangleright$ Write cached Gaussians to RAM
        **end if**
    **end while**
**end procedure**

---

## A.10 DECLARATION OF LLM USAGE

LLMs were used to polish the language in the paper, but did not write any sections on their own. Some code sections in the implementation were generated using LLMs.

