# OpenReview forum: "A LoD of Gaussians: Unified Training and Rendering for Ultra-Large Scale Reconstruction with External Memory"
_ICLR.cc/2026/Conference — Submitted to ICLR 2026_

### Official Review · Reviewer_s64K · 2025-10-20

**Soundness:** 3
**Presentation:** 2
**Contribution:** 2
**Rating:** 4
**Confidence:** 4

**Summary:**

His paper presents A LoD of Gaussians, a method that overcomes GPU memory limitations in city-scale 3D Gaussian Splatting by introducing a hierarchical LoD system combined with external memory management.

**Strengths:**

1. The core technique is novel and interesting in its details.
2. The paper includes careful analysis of memory usage.

**Weaknesses:**

1. I believe the experimental section is insufficient. First, the paper lacks detailed descriptions of the training settings. Second, the comparative experiments are incomplete—methods such as Grendel-GS and OccluGaussian, which are specifically designed for large-scale reconstruction, are not included for comparison. Moreover, the evaluation is performed on a very limited set of datasets, which restricts the generalizability of the conclusions.
2. The paper lacks references to important related work on Level of Detail (LOD) and large-scale rendering. The authors should consider citing the following works:
    - *OccluGaussian: Occlusion-Aware Gaussian Splatting for Large Scene Reconstruction and Rendering*
    - *Virtualized 3D Gaussians: Flexible Cluster-based Level-of-Detail System for Real-Time Rendering of Composed Scenes*
3. This paper presents a relatively detailed design of a 3DGS-based reconstruction framework for large-scale scenes, covering aspects from methodology to memory management. However, I find the novelty of the proposed method limited. The LoD framework is quite similar to HierarchicalGS, and the strategy of dynamically loading data from CPU to GPU has also been explored in prior work.

**Questions:**

please see the weakness above.

---

> ### Author Response · Authors · 2025-11-14
> **Response to Reviewer s64K Part 1**
>
> We are encouraged that s64K found our technique to be “novel and interesting” and thank them for their suggestions to improve the manuscript. Further, we would like to clarify a few of the points raised.
>
> *First, the paper lacks detailed descriptions of the training settings.*
>
> In addition to the main paper, 4 pages in the appendix (Sections A.4 and A.5) are dedicated to detailed descriptions of training settings. Additionally, we provide configuration files in the supplemental material for all scenes for both our and compared methods. If any details from these are missing in the appendix or main paper, we are happy to add them. Please specify what you think is missing.
>
> *Second, the comparative experiments are incomplete—methods such as Grendel-GS and OccluGaussian, which are specifically designed for large-scale reconstruction, are not included for comparison.*
>
> - GrendelGS: We specifically address our reasons for not comparing against Grendel-GS in Appendix A.5: We were unable to get access to sufficient GPU resources to run this method on datasets of this scale. (Grendel does not reduce VRAM usage in any way, meaning the entire scene has to be stored at all times) and we don’t consider methods that brute force the memory problem using multiple GPUs to be good comparison points. Instead, we opted to compare against other large-scale single GPU training methods.
>
> - OccluGaussian: OccluGaussian was only published after our paper's submission (!) and does not have an official code release at this time. The link to download their dataset is currently dead, which we already contacted the authors about 2 weeks ago. We also refer to the ICLR reviewer policy: If a paper was published (i.e., at a peer-reviewed venue) on or after July 24, 2025, authors are not required to compare their own work to that paper.
>
> *The authors should consider citing the following works: OccluGaussian and Virtualized 3D Gaussians.*
>
> We agree that these are important references and have added them in the revised manuscript. Note that both of these papers are considered "very recent work" by the ICLR reviewer guidelines and OccluGaussian was not published at time of submission.
>
> *The LoD framework is quite similar to HierarchicalGS*
>
> We agree with this assessment. The main contribution of our paper is not to introduce a new level-of-detail framework, but to extend the hierarchical LoD model such that it can be used during training and densification in conjunction with out-of-core storage using the new hierarchy cut algorithm and HSPT data structure. Note that the hierarchical LoD model is also a straight forward binary tree with distance thresholds, which is a very common approach throughout all rendering literature. Still, it is also a good fit for our purpose. Our contributions are still very distinct from Hierarchical 3DGS with us being the first approach that actually works out-of-core during training while supporting gradient flow to all levels of detail.
>
>
> *The strategy of dynamically loading data from CPU to GPU has also been explored in prior work.*
>
> Dynamically loading data from CPU to GPU is of course as old as GPUs themselves, but its applications to specifically radiance field training are underexplored in our opinion. There are several methods that explore streaming 3DGS models, but that is a substantially different task from out-of-core training, because densification, the optimizer and the drastic changes in view direction need not be accounted for.
>
> A current paper search revealed that two papers have appeared after our submission, which also focus on out-of-core training: GS-Scale (preprint) and CLM (ICCV 2025). Note that neither of these methods use level-of-detail structures, meaning their applicability is limited to scenes where frustum culling alone is sufficient to reduce the required number of loaded Gaussians and they do not contain any solutions for the problem of the entire scene being contained in the camera frustum.
>
> We would be interested to hear about any other works that we may have missed and could be considered similar to ours in this respect.

---

> ### Author Response · Authors · 2025-11-14
> **Response to Reviewer s64K Part 2**
>
> *Moreover, the evaluation is performed on a very limited set of datasets, which restricts the generalizability of the conclusions.*
>
> We would like to point out that further results are included in Section A.4, but we do agree that additional datasets would have strengthened our conclusion. Unfortunately, finding adequate datasets has been the largest challenge in putting this paper together. Our method is aimed at scenes that are both large enough that training on a single GPU is infeasible and contain sufficient variation in scales to pose challenges to the reconstruction method. Most available “large-scale” datasets fail the first hurdle, as they are maybe large in scale (covering a wide area), but shallow in detail with only a few hundred images (for comparison, MC-small-city+ contains 42.000), making them trivial to reconstruct on a single datacenter-grade or even consumer GPU.
>
> To further illustrate our point, we went through 12 different large-scale 3DGS papers and looked at all of the used datasets:
> - OccluGaussian dataset: dead link
> - MatrixCity Aerial / Street: included in the paper
> - Hierarchical 3DGS dataset: included in the paper, yet little variation as all street views
> - UrbanScene3D: Scenes contain less than a 1000 views, aerial only
> - HorizonGS dataset: Contains variation in scale, but is mostly synthetic, contains < 2000 views per scene and carefully constructs the scenes to avoid chunking artifacts by focusing only on a single building or convex structure.
> - Wayve: Street views only, less than a thousand views per scene
> - Mill19 / MegaNeRF: Included in the paper, only aerial views
> - Drone-assisted Road Gaussian Splatting with Cross-view Uncertainty: contains scale variation, but scenes easily fit onto a single GPU
> - ULSR-GS: Not publicly available
> - GauU: < 2000 images per scene, aerial only
>
> There is a severe lack of publicly-available city-scale reconstruction datasets that are not just dash-cam videos. We hope that with the curated and expanded MC-small-city+, we can contribute to setting a more solid baseline for this task.
>
> Finally, we also ask that not only the number, but also size of datasets is taken into account. The enormous scales and amounts of training data contained in these city-scale datasets should by itself help to decrease dataset bias.

---

### Official Review · Reviewer_9WiM · 2025-10-25

**Soundness:** 1
**Presentation:** 2
**Contribution:** 2
**Rating:** 0
**Confidence:** 5

**Summary:**

This paper addresses a major limitation of Gaussian Splatting: its difficulty in scaling to large, multi-scale environments (like a city with both aerial and street views). Current methods divide the scene into "chunks," which causes artifacts at boundaries, complicates training, and hits GPU memory limits. The authors propose "A LoD of Gaussians," a framework that enables the training and rendering of massive Gaussian scenes on a single consumer-grade GPU without any partitioning.

**Strengths:**

Efficient large-scale scene reconstruction and rendering are of great significance. The clarity is also fine.

**Weaknesses:**

1. The experiments are far from sufficient. The authors provide no ablations of method design and provide little illustration of the details for scaling up compared models to the large-scale scenes. So there is no guarantee about the fairness of the comparison, showing a lack of respect for the conference. Besides, the qualitative comparison is also limited to two datasets. I would strongly recommend rejecting the paper.
2. The innovation is limited. The proposed techniques are mostly a combination of existing methods or engineering efforts, making it insufficient to be accepted as an ICLR paper.

**Questions:**

See weakness.

---

> ### Author Response · Authors · 2025-11-14
> **Response to Reviewer 9WiM**
>
> We would like to respond to the comments by the reviewer, as we believe them to be inaccurate.
>
> *The authors provide no ablations of method design.*
>
> This is simply untrue, we refer to section A.3 and Table 3 for ablations on our method. If the included ablations are deemed to be insufficient, we would ask for concrete suggestions.
>
> *The authors provide little illustration of the details for scaling up compared models to the large-scale scenes.*
>
> We have interpreted this to mean that there is a lack of detail about how we run the baselines on new datasets; if this does not adequately address the concern, we kindly ask reviewer 9Wim to rephrase.
> First, we want to point out that all methods we compare against are already specialized on large-scale scenes, meaning we did not substantially change these methods to run on the reported datasets. The only changes we did introduce are detailed in A.5: “In order to train MC-small-city+ with 42.2 thousand images, it was necessary to modify CityGaussian, HorizonGS and OctreeGS to load images from disk instead of caching them in RAM or VRAM. This slows down training, but does not affect the final results.”
> Still, these methods can be very sensitive to hyperparameters, which are thoroughly documented in section A.4 and A.5. We also include full config files for all scenes and all our baselines and our own method in the supplemental material so anyone can replicate the reported results! If any of the details from these configuration files should also be included in section A.4/A.5 or the main paper, we kindly ask the reviewer for suggestions.
>
> *The qualitative comparison is also limited to two datasets.*
>
> This is also not accurate. While we include two datasets in the main paper, section A.4 provides results on additional datasets. It also demonstrates the need for more diverse datasets for city-scale 3DGS optimization, which we address in Section 5. Further, we want to point out that by their nature, an individual scene from these datasets contains more images than several “smaller scale” datasets put together.
>
> *The proposed techniques are mostly a combination of existing methods or engineering efforts.*
>
> We disagree here as well. While out-of-core techniques by nature also require significant engineering to get to run, there is significant novelty in our work:
> While H-3DGS also uses a tree structure, our approach is the first to enable true continuous, hierarchical training. There is no other approach that can propagate gradients to different scales throughout the entire training process!
> We show the very first out-of-core 3DGS training approach!
> Sequential point trees have never been adapted to 3D Gaussians or used for any caching approach.
> Although the H-SPT structure is related to point cloud literature, our unique 2-step cut structure is specifically optimized for dynamic changes, early cullling and being split across GPU and CPU.
>
> Overall, we would like to point out that from our perspective the review is lacking in constructive criticism, as the first point is inaccurate and the second overly vague. Further, the review does not make any mention of the reported results, which other reviewers are describing as “well motivated”, “strong qualitative and quantitative improvements” and “a highly impactful result”, nor lists any strength or positive beyond “fine clarity”.

---

### Official Review · Reviewer_phrf · 2025-10-28

**Soundness:** 2
**Presentation:** 3
**Contribution:** 2
**Rating:** 6
**Confidence:** 4

**Summary:**

This paper proposes A LoD of Gaussians, a unified framework for out-of-core training and rendering of ultra-large-scale 3D Gaussian Splatting scenes without spatial chunking. The core idea is to store the full scene in CPU memory, build a dynamic Level-of-Detail hierarchy, and use Hierarchical Sequential Point Trees (HSPT) to efficiently generate LoD cuts for view-dependent rendering and training. A GPU-side caching and view scheduling mechanism further reduces data transfer overhead and stabilizes training.

**Strengths:**

1. Eliminates chunk partition issues (ghosting, bleeding, merging artifacts).

2. Scales to tens of millions of Gaussians on a single GPU, a highly impactful result.

3. HSPT is a clever and elegant hybrid between hierarchy BFS and parallel SPT cutting.

4. Strong qualitative and quantitative improvements on city-scale datasets.

**Weaknesses:**

1. CPU memory remains the true bottleneck; the method is not actually hardware-light.

2. Initialization assumes good camera poses and geometry, performance may collapse otherwise.

3. The method is less advantageous when scale variation is small (single-height aerial sets).

4. Some ablation discussions are descriptive rather than analytical, more measurements would clarify causality.

5. The training speed per step is slower; the improvement is in iteration count, not iteration efficiency.

**Questions:**

1. Could the hierarchy be partially stored on SSD with async prefetch to reduce RAM pressure?

2. How sensitive is HSPT cut correctness to the surface-area-based md metric? Any cases of cut instability?

3. Does caching introduce systematic bias in reconstructed local texture, e.g., does cache reuse correlate with oversharpening?

4. Could joint pose refinement in early stages reduce reliance on precise COLMAP input?

---

> ### Author Response · Authors · 2025-11-14
> **Response to Reviewer phrf**
>
> We thank the reviewer for thoroughly engaging with our work and especially for asking insightful questions, which we are happy to answer.
>
> *CPU memory remains the true bottleneck; the method is not actually hardware-light.*
>
> Our method does require greater RAM capacities than comparable 3DGS methods. However, we only use the CPU RAM as swapping space for the GPU, whereas the challenge is mostly coming from how to design the GPU algorithm to select the right data to be streamed from the CPU. At the same time, using DirectStorage data can directly be streamed from an SSD as well (see below). Still, CPU RAM is much more affordable and accessible than the alternative of datacenter GPUs.
>
> *Initialization assumes good camera poses and geometry, performance may collapse otherwise.*
>
> As with the vast majority of Radiance Field methods, SfM initialization is required. Poor initialization will degrade performance similarly to comparable methods. For the scenes in our evaluation, initialization is already considerably noisier than on smaller datasets, due to the challenge that the scale poses to SfM methods, but the reconstruction is still successful.
> Note that our method lends itself to run SfM-free as our approach is not split into multiple phases or chunks of data, but rather allows for continuous optimizations across scales. Due to fairness of evaluation, we relied on the same initialization steps for all methods.
>
>
> *The method is less advantageous when scale variation is small (single-height aerial sets).*
>
> We agree and also mention that in the paper. We also want to emphasize on the flip side:  scale variations are very difficult or simply impossible to handle for current methods.
>
> *Some ablation discussions are descriptive rather than analytical, more measurements would clarify causality.*
>
> We are happy to provide more details. Which parts would you suggest to put more focus on and integrate into the main paper?
> **Update**: We have rewritten the ablation section and added additional measurements.
>
> *The training speed per step is slower; the improvement is in iteration count, not iteration efficiency.*
>
> Note that our approach includes streaming data from CPU memory to GPU memory during training, which allows us to handle datasets that go beyond consumer card’s memory capabilities. Obviously, this will reduce the speed of training iterations, as it includes additional memory transfer.
>
> *Could the hierarchy be partially stored on SSD with async prefetch to reduce RAM pressure?*
>
> Yes. In fact, we experimented with storing the Gaussians on an SSD instead of RAM, which reduced RAM pressure at a significant performance cost. Implementing a more sophisticated method, which stores Gaussians on disk and uses CPU RAM as an intermediate Cache is something we are looking into for future work. We also implemented a prototype for asynchronous fetching from RAM during interactive rendering, which helps reduce loading spikes.
>
> *How sensitive is HSPT cut correctness to the surface-area-based md metric? Any cases of cut instability?*
>
> We would like to ask for clarification on what constitutes HSPT cut correctness. The HSPT cut is equivalent to the BFS cut on the underlying hierarchy and is proper by definition. The only error that is introduced does not originate with the md metric, but the difference between the distance of the root and the child gaussians to the camera, which can be controlled with the size parameter. Of course, the md metric does not take viewing direction into account, as that would lead to unreasonable performance overhead, so it will not completely correlate with the number of pixels “covered” by the Gaussian.
>
> We interpret cut instability to mean instances where increasing detail level decreases quality.
> There are rare instances of cut instability, one can be seen for example in the supplemental material video “LOD_Visualization.mp4” on the fifth example. In Figure 18 of the paper, this can also be seen for some views of the MatrixCity street dataset, although the difference is minor (< 0.2 PSNR) and even non-existent or even reversed on LPIPS metric (which we generally find to be more trustworthy). In conclusion, it may occur, but is the exception rather than the norm and does not have a significant impact on final reconstruction quality.
>
> *Does caching introduce systematic bias in reconstructed local texture, e.g., does cache reuse correlate with oversharpening?*
>
> No, we have not observed that. As long as the parameter D_max is not chosen to be excessively high, the trained Gaussians are not sufficiently small in screen space to cause aliasing.
>
> *Could joint pose refinement in early stages reduce reliance on precise COLMAP input?*
>
> As mentioned before, our method lends itself to COLMAP-free approaches. It also allows information sharing over large distances due to the level-of-detail approach and thus would allow for joint pose refinement that goes beyond what chunk-based approaches would allow.

---

### Official Review · Reviewer_5wAV · 2025-11-01

**Soundness:** 3
**Presentation:** 3
**Contribution:** 3
**Rating:** 6
**Confidence:** 3

**Summary:**

This work proposes a 3DGS pipeline that avoids scene partitioning by storing the full scene out-of-core (CPU RAM) and streaming view-relevant Gaussians to the GPU.

**Strengths:**

1. Clear systems contribution for scaling 3DGS without chunking; the combination of LoD + HSPT + out-of-core streaming is well motivated and addresses real bottlenecks.
2. Practical details on hierarchy maintenance during training and scheduling/caching, with results on large multi-scale scenes.

**Weaknesses:**

1. ML novelty is limited: contributions are primarily data-structure/streaming/systems optimizations around standard 3DGS training; the learning component itself is not substantially new for ICLR.
2. While large-scale results are discussed, Lack comparisons (Training time, FPS, GPU consumption) against recent large-scene 3DGS(Octree-GS CityGaussin Momentum-GS CityGS-X).
3. Can you report sensitivity curves for cache size and LoD cut parameters vs. image quality and FPS?

**Questions:**

This paper makes significant efforts to improve the underlying rendering and training logic of 3D-GS. It would be interesting if the authors could also discuss how their approach might be applied to models like Grendel-GS, which are more sensitive to communication latency.

---

> ### Author Response · Authors · 2025-11-15
> **Response to Reviewer 5wAV Part 1**
>
> We are encouraged that 5wAV recognizes the practical improvements our method brings to scaling 3DGS. We are happy to discuss any raised questions and concerns!
>
> *Large-scale results lack comparisons (Training time, FPS, GPU consumption) against recent large-scene 3DGS(Octree-GS CityGaussin Momentum-GS CityGS-X).*
>
> There are multiple aspects to this point:
> - CityGS-X has only been published at ICCV2025 and thus cannot be considered prior work according to the reviewer guidelines. Furthermore, there is no real-time viewer available. We are happy to cite the work and discuss it.
> - Momentum-GS has also only been published at ICCV2025. Also, there is no real-time viewer available. We are happy to cite the work and discuss it.
>
> We do provide GPU consumption metrics (we assume you refer to memory) for Octree-GS and CityGaussian in Table 1.
>
> We did not include training time for the different methods, as many of them were pushing the limits of GPU memory and thus training images had to be loaded from disk, where the methods would usually cache them. This step may adversely affect the training times. We are still happy to provide approximate training time from our logs here (for MC-small-city+):
> - Ours: 35h  for 136.7M Gaussians
> - HorizonGS: 85h
> - Hierarchical 3DGS: 70h for 83.1M Gaussians (Note that this was for training only a third of the dataset to avoid OOM during eval)
> - CityGaussian: 8h for 2.7 M Gaussians
>
> HorizonGS had the longest time to optimize a single chunk (2.5 hours), while the splitting heuristics of H-3DGS produced the greatest number of chunks (97). CityGaussian takes the lead here, as it uses significantly less training iterations per chunk than the other methods. But it discards the vast majority of densified Gaussians during merging to avoid artifacts.
>
> FPS is also difficult to assess, as the approaches do not provide efficient (working) viewer implementations. CityGaussian requires all Gaussians to reside in GPU memory and thus no data transfer from the CPU is carried out. When sufficient VRAM is available, CityGaussian achieves high FPS (60+), because it does not use any streaming and only a light-weight LoD method.  Hierarchical 3DGS claims speeds of around 30 FPS for their own datasets on A100 GPUs. We were unfortunately not able to replicate or expand these results as even with significant effort, we could not get the viewer running on any of 4 different machines, which is a problem also documented by numerous GitHub issues. The viewer included with HorizonGS is simply non-functional, as admitted by the authors on GitHub. The test images for evaluation rendered at less than 2 FPS on an H200 due to the overhead of the used ScaffoldGS method. Our rendering time can be found in the appendix, demonstrating that we can stream data with interactive framerates of 12-20 fps depending on the scene using an RTX 3090 GPU.
>
> We would be happy to put some of this information into the main paper.
>
>
>
> *ML novelty is limited: contributions are primarily data-structure/streaming/systems optimizations around standard 3DGS training; the learning component itself is not substantially new for ICLR.*
>
> We partially disagree here. It is true that the focus of our work lies on data structures and enabling large scale 3DGS training. However, we also are the first to enable training across scales, where gradients are applied at different scales throughout the entire training process.
> Also note that “infrastructure, software libraries, hardware, etc.” is specifically mentioned in the call for papers.
>
>
>
> *Can you report sensitivity curves for cache size and LoD cut parameters vs. image quality and FPS?*
>
> Sensitivity curves for LoD level and image quality are included in Figure 18 of the appendix. We have added sensitivity curves for FPS in Figure 18 and Cache Size / FPS sensitivity curves in Figure 19 of the revised document.
>
> We would like to ask for clarification with regards to the Cache Size / Quality curves. While cache size would have some effect on the output frames, we are unsure how to report quality metrics without available ground truth on the camera paths. If this is instead referring to the cache size during training, our preliminary experiments reported in A.3 have shown no discernible impact of caching on final reconstruction quality. We could add more data points to support this claim, but the required training runs would take a bit of time to complete.

---

> ### Author Response · Authors · 2025-11-15
> **Response to Reviewer 5wAV Part 2**
>
> *It would be interesting if the authors could also discuss how their approach might be applied to models like Grendel-GS, which are more sensitive to communication latency.*
>
> We don’t think that there is a useful way to apply our method to Grendel-GS. If our method was used for multi-GPU training, it makes more sense that each GPU performs individual training iterations and then writes the results back to a common storage instead of the Grendel Method of distributing one iteration among different GPUs. Grendel-GS is based on the assumption that there is sufficient VRAM (over multiple GPUs) to store and train the entire scene at highest detail, while our method assumes the opposite. This leads to drastically different design decisions, which we do not believe to be compatible.

---

> > ### Comment · Reviewer_5wAV · 2025-11-27
> >
> > Thanks for the response and clarification. It seems that this paper proposes an alternative solution for large-scale scene reconstruction, distinct from partition-based and parallel-based methods, which is a novel attempt.
> > However, my major concern is the performance of this paper.
> >
> > - **Partition Elimination**: This method eliminates the partition strategy compared to CityGaussian. However, partitioning has already been alleviated in parallel-based methods such as Grendel-GS and CityGS-X. Notably, Grendel-GS is an ICLR'25 paper, so this line of work has been explored for about a year already.
> > - **Real-time Rendering**: This method is built on the assumption that even multiple GPUs are still insufficient for real-time rendering of an entire large-scale scene. However, this assumption is not empirically supported in the paper and is unlikely to hold in practical scenarios, since parallel-based methods can leverage multiple machines to achieve both training and rendering.
> > - **Training Time**: According to the rebuttal, the proposed method requires significantly more training time than CityGS (35h v.s. 8h), which is even less competitive when compared to recent parallel-based approaches.
> > - **Render Quality**: According to the experiments in Table 5 of the appendix, the rendering quality is worse than CityGS on several common scenes (e.g., *Rubble*, *Building*), which suggests that the applicability of the proposed method is quite limited.
> >
> > In summary, while the reviewer acknowledges the novelty of the proposed method, its performance remains a major concern. Hence, the reviewer prefers to keep the score unchanged.

---

> ### Author Response · Authors · 2025-11-27
> **Response to Comment by Reviewer 5wav**
>
> We sincerely thank the reviewer for taking the time to consider and respond to our rebuttal and would like to provide some more context to the data points included in the reponse:
>
> (Note that figure numbers refer to the revised manuscript)
>
> *Partitioning has already been alleviated in parallel-based methods such as Grendel-GS and CityGS-X*
>
> This is correct, but these methods come at great costs. For example, for training just a small part of the MatrixCity dataset (composing only an 8th of the total number of images), CityGS-X uses 4 RTX 4090 and Grendel-GS even uses 16 A100 GPUs (coming to a conservative cost estimate of 160,000$ for the required hardware). This is compared to our requirements of a single RTX 3090 for the *entire* MatrixCity scene, where we train up to 150 million Gaussians compared to Grendel’s 24 million. We remain firm that these differences are very significant and that our method scales drastically better in resource requirements compared to parallel-based methods.
>
> *This method is built on the assumption that even multiple GPUs are still insufficient for real-time rendering of an entire large-scale scene.*
>
> This is not at all something we want to claim or support. For example, the prior work RetinaGS already demonstrated 3DGS training on 64xA100 GPUs in parallel with an astonishing combined 5 TB of VRAM. Against resources of this scale, our method could never compete on quality or speed. The aim of our method is to enable training of huge scenes on hardware that is cheaper by orders of magnitude and could not otherwise support such training (or only with severe scene division artifacts).
>
> *The proposed method requires significantly more training time than CityGS (35h v.s. 8h)*
>
> We would like to point out that the result of CityGS training on that scene shows a severe degradation of quality compared to our result (We suggest comparing the results in Figure 7, we can provide additional data if this is not convincing). We will run experiments to prove that our method can match and exceed the quality metrics of CityGaussian within 8 hours of training by reducing the number of iterations.
>
>
> *According to the experiments in Table 5 of the appendix, the rendering quality is worse than CityGS on several common scenes (e.g., Rubble, Building), which suggests that the applicability of the proposed method is quite limited.*
>
> These common scenes were included in the paper Mega-NeRF, which is based on divide-and-conquer. For this reason, the camera poses were chosen to complement the strengths of that approach. All views are captured at equal height, pointing downward onto flat terrain with little co-visibility between cameras (Please compare the visualization of the camera poses of rubble in Figure 21 with the camera poses of MatrixCity in Figure 20). As explained in the paper, for datasets with these specific conditions, divide-and-conquer based methods are the superior choice, because the limited co-visibility negates chunking artifacts. We would argue that this is a special case where the dataset has to be especially tailored to the reconstruction method and that naturally capturing a dataset would lead to much more complex patterns of camera co-visibility.
>
> There is a cost of visual quality from the out-of-core training and LoD model compared to full in-core training that can be seen in the Mill19 results. For scenes with less structured camera poses (introducing variations in view angles and scales), this cost is significantly outweighed by the quality lost from chunking artifacts, as demonstrated on MatrixCity for example. We would argue that not the applicability of our method is limited, but that the divide-and-conquer approach is limited in that it degrades severely when camera poses do not follow specific patterns, while our approach does not.
>
> **In Summary**: Previously, large-scale reconstruction was possible either using expensive computing clusters or by using divide-and-conquer methods, which produce severe artifacts when the camera poses are not carefully controlled and require much more VRAM to render the entire scene in one piece when training is finished. Our method aims to provide a third option, that can train and render the finished model *on consumer hardware*, but does not suffer from chunk artifacts.

---

### Author Response · Authors · 2025-11-21
**Revised Paper Uploaded and Request for Feedback**

We thank the reviewers for their comments and have aimed to address their concerns with the revision to the manuscript. Here is an overview of the changes in the revision:
- Rewrote the ablations chapter to be more analytical by providing more measurements (Section A.3)
- Added FPS/LoD sensitivity curves (Section A.3)
- Added FPS/Cache Size sensitivity curves (Section A.3)
- Added Citations for OccluGaussian and Virtualized 3D Gaussians (Section 2)
- Moved some details for scaling up compared works from appendix to main paper
- Moved Figure 1 from appendix to main paper
- Fixed Typos

We have also posted detailed responses to each of your individual reviews and would greatly value your feedback on these updates to ensure a productive discussion. If there are any remaining issues preventing you from reconsidering your score, please let us know so we can address them before the revision deadline on 3.12.

---

### Meta-Review · Area_Chair_wzxb · 2026-01-03

**Summary:**

I have carefully reviewed the comments from each reviewer and have also engaged in in-depth discussions with the reviewer（9WiM） who gave a score of 0 to understand his/her motivations for the rating. While most reviewers acknowledged the value of this work to the community, they expressed significant concerns regarding the insufficiency and lack of rigor in the experiments. During my communication with Reviewer 9WiM, he/she considered this work to be a good Tech Report; however, in the context of ICLR standards, they found the methodological innovation, as well as the rigor and completeness of the paper and experiments, to be inadequate. Personally, I agree with it. Therefore, the authors need not take this reviewer's feedback personally.

I also further exchange with other reviewers. The reviewers acknowledge the paper's novelty in proposing a non-partitioned, non-parallel alternative for large-scale reconstruction. However, they still raise major concerns about performance: the claimed advantage of partition elimination is not compelling given recent parallel methods, the method underperforms in both training time (35h vs. 8h) and rendering quality on several scenes(Rubble and Building) compared to baselines.

In summary, this paper is a solid work with in-depth engineering designs. However, there is significant room for improvement in terms of writing (including the organization of content between the main text and supplementary materials) and experimental validation. I believe that with further refinement of the work, particularly if experiments and comparisons can be extended to more large-scale or extremely large-scale scenarios(You can't just ignore related methods like Grendel GS and etc., and claim you only care about using worse devices for reconstruction.), it has the potential to become an excellent contribution. Nevertheless, I maintain a "reject" decision for the current version.

**Reviewer Concerns:**

Please see above.

**Reviewer Scores:**

Please see above.

---

### Decision · Program_Chairs · 2026-01-26

Reject